# Hypersensitive intercellular responses of endometrial stromal cells drive invasion in endometriosis

Chun-Wei Chen[1†], Jeffery B Chavez[1†], Ritikaa Kumar[1], Virginia Arlene Go[2], Ahvani Pant[1], Anushka Jain[1], Srikanth R Polusani[1], Matthew J Hart[3], Randal D Robinson[2], Maria Gaczynska[4], Pawel Osmulski[4], Nameer B Kirma[4]*, Bruce J Nicholson[1]*

[1]Department of Biochemistry and Structural Biology, UT Health San Antonio, San Antonio, United States; [2]Department of Obstetrics and Gynecology, UT Health San Antonio, San Antonio, United States; [3]Center for Innovative Drug Discovery, UT Health San Antonio, San Antonio, United States; [4]Department of Molecular Medicine, UT Health San Antonio, San Antonio, United States

*For correspondence:
kirma@uthscsa.edu (NBK);
nicholsonb@uthscsa.edu (BJN)

†These authors contributed
equally to this work

Competing interest: See page
18

Reviewing Editor: Hugh Taylor, ,
United States

**Abstract** Endometriosis is a debilitating disease affecting 190 million women worldwide and the greatest single contributor to infertility. The most broadly accepted etiology is that uterine endometrial cells retrogradely enter the peritoneum during menses, and implant and form invasive lesions in a process analogous to cancer metastasis. However, over 90% of women suffer retrograde menstruation, but only 10% develop endometriosis, and debate continues as to whether the underlying defect is endometrial or peritoneal. Processes implicated in invasion include: enhanced motility; adhesion to, and formation of gap junctions with, the target tissue. Endometrial stromal (ESCs) from 22 endometriosis patients at different disease stages show much greater invasiveness across mesothelial (or endothelial) monolayers than ESCs from 22 control subjects, which is further enhanced by the presence of EECs. This is due to the enhanced responsiveness of endometriosis ESCs to the mesothelium, which induces migration and gap junction coupling. ESC-PMC gap junction coupling is shown to be required for invasion, while coupling between PMCs enhances mesothelial barrier breakdown.

## Editor's evaluation

The authors show how gap junction proteins are induced to be expressed at the cell surface when the endometrium interacts with the mesothelium, and that this induction is much higher in endometrial stromal cells from endometriosis patients than controls. Strengths include the use of various methods, with useful results demonstrating that gap junction coupling between endometrial stromal cells and the mesothelium is required for invasion in vitro. These findings provide solid support for the role of the endometrium in allowing endometriosis to be an invasive disorder.

## Introduction

Endometriosis is a chronic inflammatory disease affecting ~10% of reproductive-age women, or 190 million worldwide (*Shafrir et al., 2018*), characterized by the presence of endometrial tissue in extrauterine lesions on the pelvic peritoneum, ovary and bowel surface. In the US, endometriosis is diagnosed in 35–50% of women with pelvic pain, and up to 50% of women with unexplained infertility (*Rogers et al., 2009*). Reliable diagnosis requires invasive abdominal surgery, resulting in an average

delay of 6.7 y from symptom onset to diagnosis, with 2/3rds of patients being misdiagnosed at some point (*Bontempo and Mikesell, 2020*; *Mettler et al., 2003*). Thus, the disease imposes a significant socioeconomic burden of ~$80 billion per year for the US alone between prolonged healthcare costs, and loss of productivity (*Soliman et al., 2016*). Treatment of the disease is also limited to excision of lesions, which usually return, or hormonal management of pain, which only further compromises fertility.

The most widely accepted model for the pathogenesis of endometriosis is retrograde menstruation, in which sloughed endometrial tissue during menses traverses the fallopian tubes (or perhaps in rare cases enters the lymphatic or blood circulation) and when deposited in the peritoneal cavity, forms invasive lesions that remain sensitive to hormonal cycles (*Sampson, 1927*). Other origins of ectopic endometrial lesions, including Mullerian remnants, metaplasia of coelomic stem cells, or endometrial stem cells have also been proposed (*Lauchlan, 1972*; *Missmer et al., 2004*; *Sasson and Taylor, 2008*), which can explain endometriosis in the absence of menstruation, clonal similarities in ectopic lesions and rare male endometriosis. However, the preponderance of evidence indicates that lesions are of endometrial origin (reviewed in *Burney and Giudice, 2012*). A major outstanding question is why retrograde menstruation, which is estimated to occur in up to 90% of women, would only result in endometriosis in 10% of women (*Burney and Giudice, 2012*). One explanation is that there are specific factors that predispose patients to disease development, but it remains unresolved as to whether these lie in the uterus (the 'seed') or the peritoneum (the 'soil'). Several studies have reported molecular differences in the eutopic endometrium of women with endometriosis (*Burney et al., 2007*; *Rogers et al., 2009*; *Ulukus et al., 2006*; *Yu et al., 2014*; *Lin et al., 2022*), including enhanced survival (*Jones et al., 1998*) and invasive potential (*Lucidi et al., 2005*) that could promote lesion formation in the pelvic cavity (*Guo et al., 2004*; *Hastings and Fazleabas, 2006*; *Tamaresis et al., 2014*). But changes in peritoneal factors can also contribute, including hormonal environment (*Parente Barbosa et al., 2011*), oxidative stress, inflammation (*Augoulea et al., 2012*), and decreased immune clearance (*Oosterlynck et al., 1991*). The problem remains to distinguish which of these changes are consequences as opposed to causes of the disease, an issue that requires a greater understanding of invasive mechanisms.

While much work has been done on the consequences of endometriosis in terms of inflammation, hormone responsiveness, and impact on fertility, little has focused on the initial causes of lesion formation. We know from other invasive processes like metastasis, that such behavior requires enhanced migratory behavior, typically after an epithelial to mesenchymal transition, followed by contact-mediated intercellular interactions between the invading and target tissues. These involve initial adhesion and subsequent gap junction formation that has been proposed to trigger disruption of the barrier functions of the target tissue, although understanding of specific mechanisms are still limited. Gap junctions, composed of connexin (Cx) proteins encoded by a family of 21 GJ(A-D) genes, mediate direct contact and communication between most cells of the body via the exchange of ions as well as metabolites and signaling molecules <1 kD (*Goldberg et al., 1999*; *Weber et al., 2004*; *Hernandez et al., 2007*).

Gap Junctions have been implicated in other invasive processes, like metastasis. In a global screen of cervical squamous carcinoma, Cx43 emerged as one of three genes (along with PDGFRA2 and CAV-1) central to cancer invasion and metastasis (*Cheng et al., 2015*). Interestingly, expression of functional gap junctions is suppressed in most primary tumors, as they suppress growth, But significant induction of Cx43 and/or Cx26 gap junctions, either by increased expression or trafficking to the cell surface (*Kanczuga-Koda et al., 2006*), has also been associated with metastatic breast cancer (*Naoi et al., 2007*; *Stoletov et al., 2013*), prostate cancer (*Zhang et al., 2015*; *Lamiche et al., 2012*), and melanoma (*Ito et al., 2000*). GJs appear to exert their effects both during intravasation and extravasation (*el-Sabban and Pauli, 1991*; *el-Sabban and Pauli, 1994*; *Ito et al., 2000*; *Naoi et al., 2007*), as well as forming hetero-cellular GJIC with the target tissue that pass miRNAs or cGAMP to promote target receptivity and an inflammatory environment (*Lamiche et al., 2012*; *Hong et al., 2015*; *Chen et al., 2016a*).

Gap junctions have also been associated with multiple aspects of the other major pathology of endometriosis, infertility (*Nair et al., 2011*; *Winterhager and Kidder, 2015*), and GJs have also been shown to be involved from the earliest phases of oocyte meiosis (*Simon et al., 1997*; *Richard and Baltz, 2014*) to endometrial decidualization (*Kaushik et al., 2020*), blastocyst implantation (*Grümmer*

*et al., 1996*: *Diao et al., 2013*), and vascularization of the endometrium during pregnancy (*Laws et al., 2008*).

Despite these links between GJs and the two major pathologies of endometriosis, studies have been limited to tracking connexin expression. Immunocytochemistry showed a shift in Cx expression of endometrial epithelial cells (EECs) from primarily Cx26 (*GJB2*) with some Cx32 (*GJB1*) in the uterus (*Jahn et al., 1995*), to Cx43 (*GJA1*) in peritoneal (ectopic) endometriotic lesions (*Regidor et al., 1997*). A similar switch in EEC Cx expression profile was reported ectopically in the uteri of baboons with endometriosis (*Winterhager et al., 2009*), but this was not seen in human patient samples where Cx expression of EECs remained unaltered (*Yu et al., 2014*). In contrast, ESCs have been reported to retain Cx43 expression in both eutopic and ectopic locations, although at significantly reduced levels in endometriosis patients (*Nair et al., 2008*, *Yu et al., 2014*). The reduced Cx43 expression in the uterus has been suggested to contribute to infertility associated with endometriosis (*Yu et al., 2014*), but to date, no studies have explored the role of GJs in lesion formation. Of particular relevance to this is the consistent observations, mentioned above, that connexin expression is repressed in primary tumors, yet is induced in metastatic tumor cells to promote invasion. We investigate this same connection here with regard to endometriosis using primary endometrial stromal and epithelial cells isolated from 22 control and 22 endometriosis patients from stages I-II and III-IV of the disease (*Table 1*).

## Results

### ESC and EEC mixes from endometriosis patients are more invasive, with ESCs being the primary invaders

As the first step in lesion formation, or any invasive process, is adhesion to the target, we tested the two major endometrial cell types for adhesiveness to mesothelial cells, as this would indicate which cell type we should focus on as the primary instigator of invasion. Peritoneal Mesothelial Cells (PMCs, specifically the LP9 cell line) or primary EECs or ESCs isolated from patients as described in Methods (see *Figure 1—figure supplement 1*) were attached to the cantilever of an Atomic Force Microscope (AFM) and brought into contact with LP9 mesothelial cells in a monolayer, and after 30 s, the force needed to separate the cells was measured (*Figure 1A - Sancho et al., 2017*; *Roca-Cusachs et al., 2017*). PMCs show low levels of adhesion to one another and to EECs, but sixfold greater forces were needed to separate ESCs from PMCs (*Figure 1B*). These measurements were conducted with cells from control patients. ESCs from endometriosis patients showed even higher levels of adhesion, as they were difficult to separate from PMCs even after only 1–2 s of contact, precluding accurate measurement of force using our instrumentation.

We then directly assessed invasiveness using an established 3D-invasion model (*Nair et al., 2008*). Endometrial cells labeled with a lipophilic fluorescent dye (Di-O) are dropped onto a hormone depleted Matrigel-coated Boyden chamber on which is grown a confluent monolayer of the LP9 PMCs (*Figure 1C*). Neither ESCs nor EECs invaded through the membrane alone, confirming a dependence on a PMC monolayer for invasion. PMCs alone showed limited invasiveness, but this was excluded as only Di-O labeled cells were counted. Comparisons of ESC and EEC invasiveness from all patients showed ESCs to be twofold more invasive (*Figure 1D*), consistent with their higher level of adhesiveness to PMCs observed above. This led us to focus our invasion comparisons between control (8) and endometriosis (11) patients primarily on ESCs. ESCs from endometriosis patients were 4–6 fold more invasive than from controls (*Figure 1E*) mostly due to patients from more advanced disease (*Figure 1 - support data 1*). This difference was less (~2.5 fold) when invasion was measured in the absence of a serum gradient, or at higher gradients (data not shown), indicating that endometriosis ESCs are more responsive to low-level chemo-attractant gradients. We also observed a similar difference of ~ fourfold when we compared invasiveness across PMCs derived from either control or endometriosis patients (*Figure 1E*).

In the retrograde model of endometriosis, fragments of the endometrium, containing both ESCs and EECs, invade the mesothelium, explaining why both cell types are found in endometriotic lesions. So while EECs alone were less invasive, we did test the effect of mixed ESC/EEC cultures in invasion across PMCs, but in the absence of an attractive serum gradient to more closely mimic in vivo conditions. When EECs were mixed in equal numbers with ESCs they enhanced invasion by 1.5-fold in control samples (n=9), and 2.1-fold in endometriosis samples (n=8), but this was only significant

**Table 1.** Patient data.

| Patient | Ethnicity | Age | BMI |
|---|---|---|---|
| CONTROL | | | |
| 1 | Cauc | 22 | 18.1 |
| 9 | NS | 35 | 27 |
| 10 | Cauc | 36 | 31 |
| 12 | Hisp | 38 | 30 |
| 14 | Hisp | 40 | 29.2 |
| 17 | Cauc | 23 | 32.6 |
| 21 | Hisp | 25 | 28.3 |
| 25 | Afr Am | 33 | 25 |
| 30 | Afr Am/Hisp | 37 | 43.4 |
| 33 * | Hisp | 31 | 39.6 |
| 34 | Cauc | 30 | 38.2 |
| 36 | Cauc | 36 | 33.1 |
| 37 * | Cauc | 28 | 37.3 |
| 38 * | Hisp | 25 | 27.7 |
| 45 * | Cauc | 33 | 38 |
| 47 * | Hisp | 29 | 24.5 |
| H11 | Hisp | 38 | 34 |
| H19 | Cauc | 25 | 28 |
| H20 | Cauc | 45 | 25 |
| H25 | Hisp | 26 | 29 |
| H27 | Cauc | 26 | 19 |
| H47 | Cauc | 24 | 33 |
| ENDOMETRIOSIS I-II | | | |
| 4 | Hisp | 35 | 22.5 |
| 16 | Pac Isl | 30 | 28 |
| 23 | Cauc | 25 | 27.4 |
| 24 | Cauc | 35 | 27.6 |
| 26 | Cauc | 24 | 23.1 |
| 27 | Cauc | 31 | 31.3 |
| 31 | Hisp | 30 | 25.8 |
| 32 | Cauc | 39 | 29.9 |
| 35 * | Cuac | 28 | 21.7 |
| 39 * | Hisp/Pac Isl | 25 | 24.2 |
| 43 | Cauc | 25 | 40.2 |
| ENDOMETRIOSIS III-IV | | | |
| 2 | Cauc | 26 | |
| 3 | Cauc | 31 | |
| 5 | Afr Am | 28 | 18.9 |

*Table 1 continued on next page*

*Table 1 continued*

| Patient | Ethnicity | Age | BMI |
|---|---|---|---|
| 6 | Cauc | 41 | 45.1 |
| 7 | Hisp | 40 | 20.3 |
| 13 | Cauc | 37 | 23 |
| 15 | Cauc | 23 | 25 |
| 19 | Cauc | 30 | 22 |
| 40 * | Hisp | 34 | 18.9 |
| 41 | Cauc | 32 | 23 |
| 42 | Cauc | 34 | 21 |

OCP, Oral contraceptives; IUD, Intra-uterine device; ES, Early secretory; MS, Mid-secretory; LS, Late Secretory; M, Menstruation.
*PMCs also obtained.

(p<0.01) in endometriosis samples (*Figure 1F*). Differential labeling the two cell types (*Figure 1—figure supplement 2*) confirmed that ESCs were the primary invading cells, with EECs making up 20% and 40% of the invading cells in controls and endometriosis, respectively (*Figure 1—source data 1*).

While most endometriosis lesions are restricted to the peritoneal cavity, some (<5%) can be found outside, even as far as the lungs and brain. Such lesions clearly cannot arise form retrograde menstruation, but as we have shown that ESCs from patients are highly invasive through PMCs, we tested whether they may also be able to intravasate into the circulation, allowing further spread. A comparison of eight patients with different invasive tendencies demonstrated that invasiveness across a PMC monolayer was highly correlated with invasiveness across an endothelial cell monolayer of HUVECs (*Figure 1G*), suggesting that spread on endometrial cells through the circulation may also be enhanced in endometriosis.

## ESCs from endometriosis patients show greater inherent motility, and this is further enhanced by PMCs

To assess another major contributor to invasiveness, we compared the motility of ESCs from control and endometriosis patients using a wound healing assay (*Figure 2A and B*). Comparisons of ESCs from 15 control and 11 endometriosis patients (4 stages I-II and 7 stages III-IV) revealed a twofold increase in motility associated with disease (*Figure 2C*). In a subset of these patient samples (eight controls and four endometriosis patients) we also tested the effects of co-culture of ESCs with LP9 PMCs. The motility of co-cultures compared to ESCs grown alone was not significantly changed in control samples but increased by 1.5-fold in endometriosis samples (*Figure 2D*). We could not assess the effect of EECs on ESC motility due to low adhesiveness of EECs, leading to selective loss of these cells and disruption of the cell monolayer needed for motility measurements. While these data show a net threefold difference in motility between control and endometriosis ESCs in the presence of PMCs, this does not fully account for the sixfold difference in invasiveness, indicating that additional factors play a role.

## PMCs induce gap junction intercellular coupling (GJIC) with ESCs

Adhesion, motility, and invasive processes have all been shown to be regulated at some level by gap junctions, either between like cells or between invading cells and the target tissue. To explore the potential role of GJs in endometriosis, we utilized an automated variant of the 'parachute' technique to measure GJIC as the rate of spread of preloaded calcein dye from donor cells (D) dropped onto a monolayer of acceptor cells (A) (*Figure 3A*). While we had observed some changes in the expression of gap junction genes in ESCs and EECs with endometriosis (*Chen et al., 2021*), we saw only modest changes (<35%) in the most highly expressed isoform, Cx43 or in homo-cellular GJIC between ESCs (*Figure 3B*) between control (black) and endometriosis (gray) patients. However, hetero-cellular GJIC between ESCs and PMCs, as would occur at the onset of lesion formation, was induced at higher

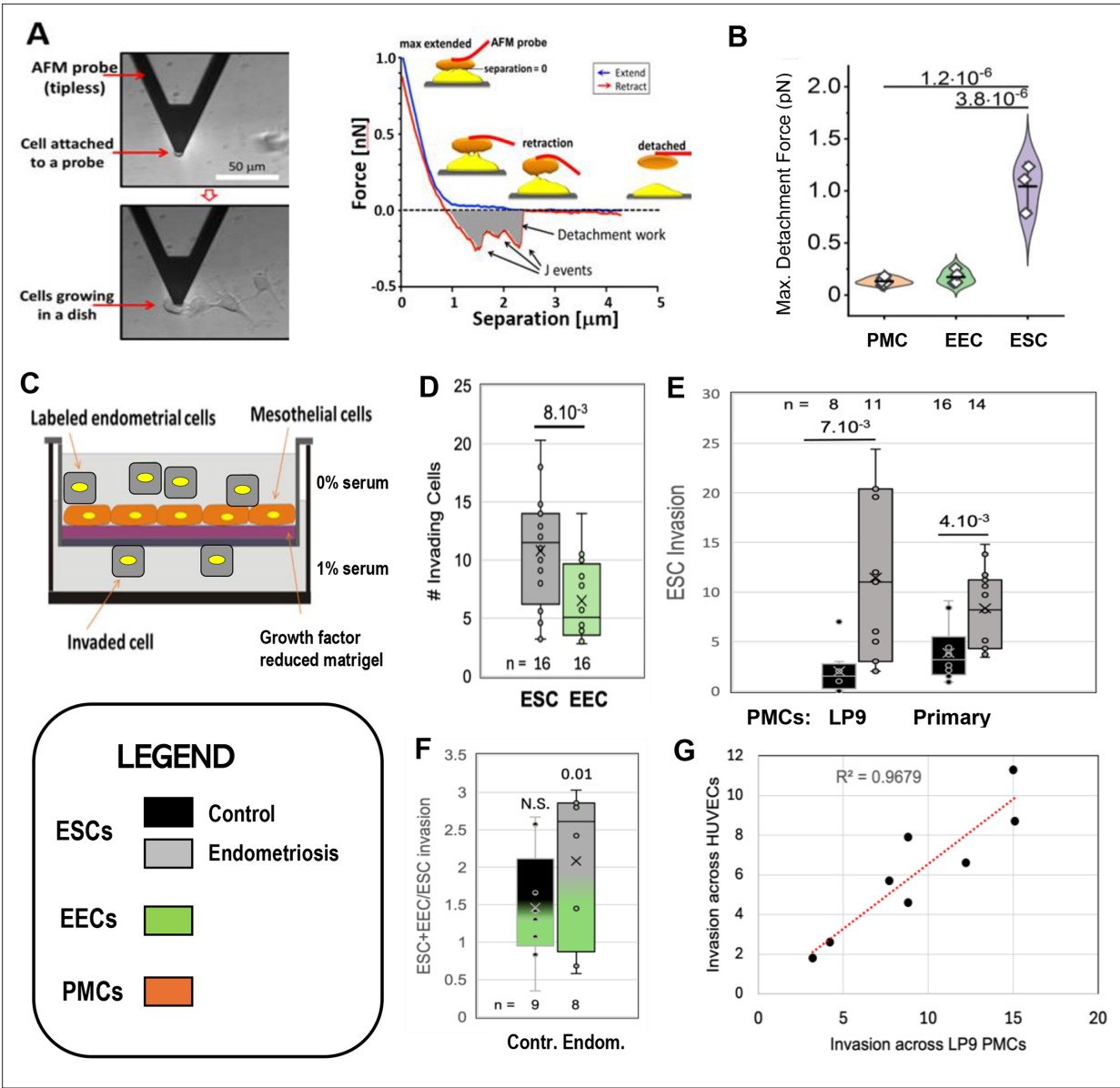

**Figure 1.** Characterization of endometrial cells from control and endometriosis patients. Adhesiveness. (**A**) The force needed to separate a cell attached to an atomic force microscope (AFM) cantilever tip (left) from peritoneal mesothelial cells (PMCs) growing on a dish (left) was calculated from a force/distance curve (right). (**B**) LP9 PMCs show similar adhesion to one another as to endometrial epithelial cells (EECs), but much stronger adhesion to endometrial stromal cells (ESCs) (3–6 technical replicates). Invasiveness. (**C**) A 3-D ex-vivo model measured endometrial cell invasion across a PMC monolayer in a Boyden chamber. (**D**) Consistent with their lower adhesion, EECs were twofold less invasive than ESCs across all patients. (**E**) ESCs from Endometriosis patients (n=7) were more invasive than those from controls (n=6), both through an LP9 PMCs (> fivefold difference) or primary PMCs (fourfold difference) derived from both control (n=5) or endometriosis (n=3) patients. (**F**) Mixes (1:1) of ESCs and EECs from the same control (n=6) or endometriosis (n=7) patients were 1.5 and 2.1-fold more invasive, respectively than ESCs alone (significance values represent the difference of co-cultures from ESCs alone). (**G**) Invasion of ESCs from eight patients across LP9 PMC or HUVEC monolayers were highly correlated. Number of repeats for each condition in D - F is shown. Significance based on two-tailed t-test. Full data in *Figure 1—source data 1*. Legend applies to *Figures 1–4*.

The online version of this article includes the following source data and figure supplement(s) for figure 1:

**Source data 1.** Data tables, and statistical analyses, for *Figure 1B, E and F*.

**Figure supplement 1.** Immunocytochemical assessment of epithelial and stromal cell isolations from patients.

**Figure supplement 2.** Relative invasiveness of endometrial stromal cells (ESCs) and endometrial epithelial cells (EECs).

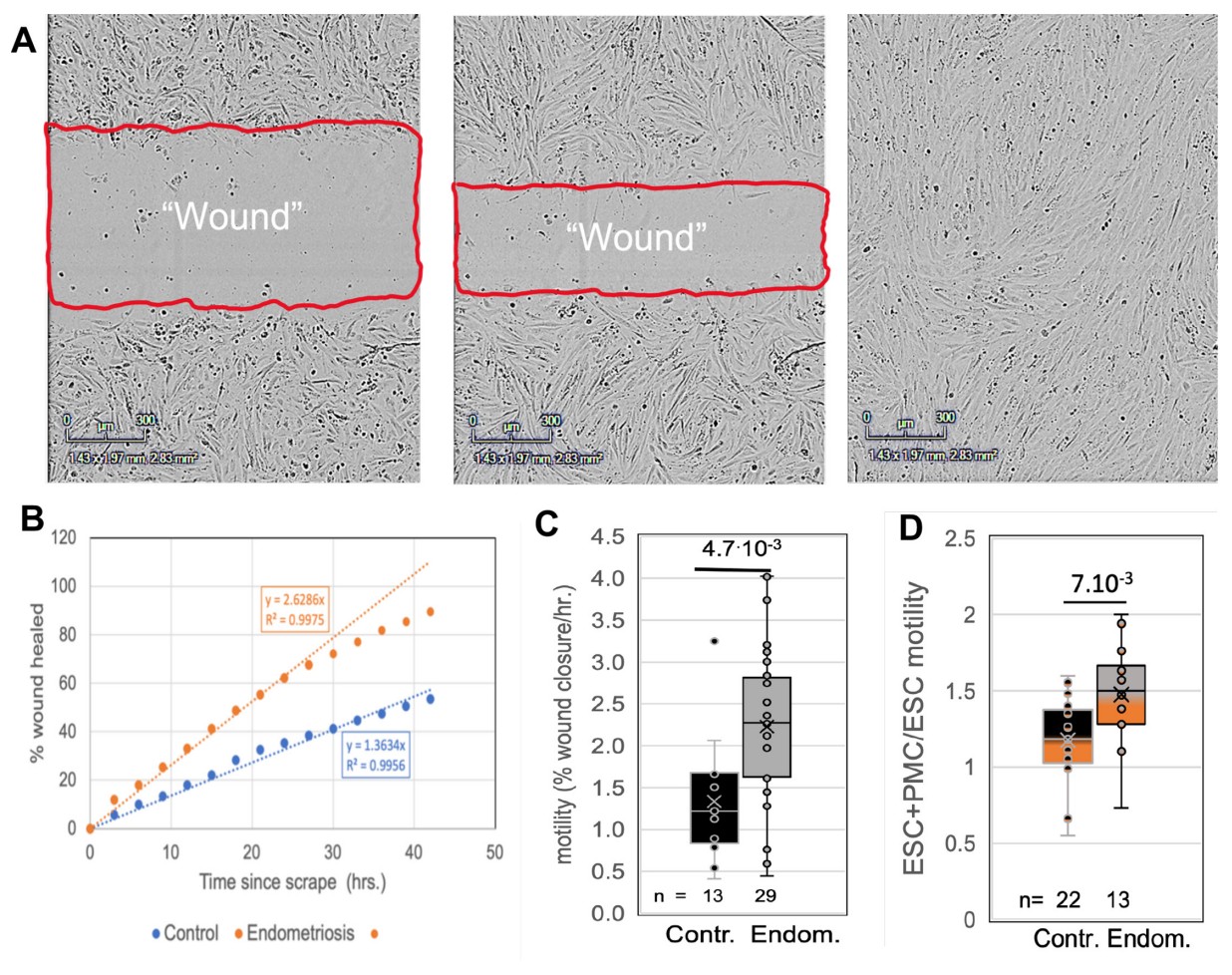

**Figure 2.** Comparisons of motility of patient endometrial stromal cells (ESCs). (**A**) Motility was measured by rates of wound clousre in in an incucyte system (images at 0, 24 and 48 hr after scraping). (**B**) Motility is measured by fits to the linear portion of the wound closure over time. (**C**) ESCs from endometriosis patients (n=10) show higher the motility than from control patients (n=9). (**D**) Mixing LP9 peritoneal mesothelial cell s (PMCs) with ESCs further increases motility of Endometriosis ESCs, while little effect is seen in control ESCs. The number of repeats for each condition is shown. Significance based on two-tailed t-tests. Full data in *Figure 2—source data 1*.

The online version of this article includes the following source data for figure 2:

**Source data 1.** Original data tables and statistical analyses for *Figure 2C and D*.

levels as disease progressed (*Figure 3C*), from twofold in controls to over threefold in stage III-IV endometriosis (p<0.01).

Since the induction of GJIC was observed within the 2- hr timespan of our coupling assay, it seemed likely it was not due to transcriptional activation. Thus, we examined the distribution of Cx43 protein in ESCs before and after contact with PMCs (*Figure 3D–K*). ESC cultures alone show most of the Cx43 staining is intracellular, particularly evident in non-confocal images (*Figure 3D–E*). Some punctate staining at cell-cell interfaces indicative of gap junction plaques (arrowheads) is evident, particularly in confocal images (*Figure 3F–G*). By contrast, in mixed ESC/PMC cultures, there is much reduced intracellular staining within most, although not all, ESCs (green * labeled cells in *Figure 3H,I and K*). Punctate staining at cell-cell interfaces is now more frequently observed between ESCs and PMCs (solid yellow arrowheads), as well as between ESCs (green-filled yellow arrowheads) (*Figure 3H and J*). These gap junction plaques are often found on PMC processes that cross ESC cell bodies (*Figure 3K*), or the reverse, and are far more frequent in ESC-PMC co-cultures than in ESC homocellular cultures.

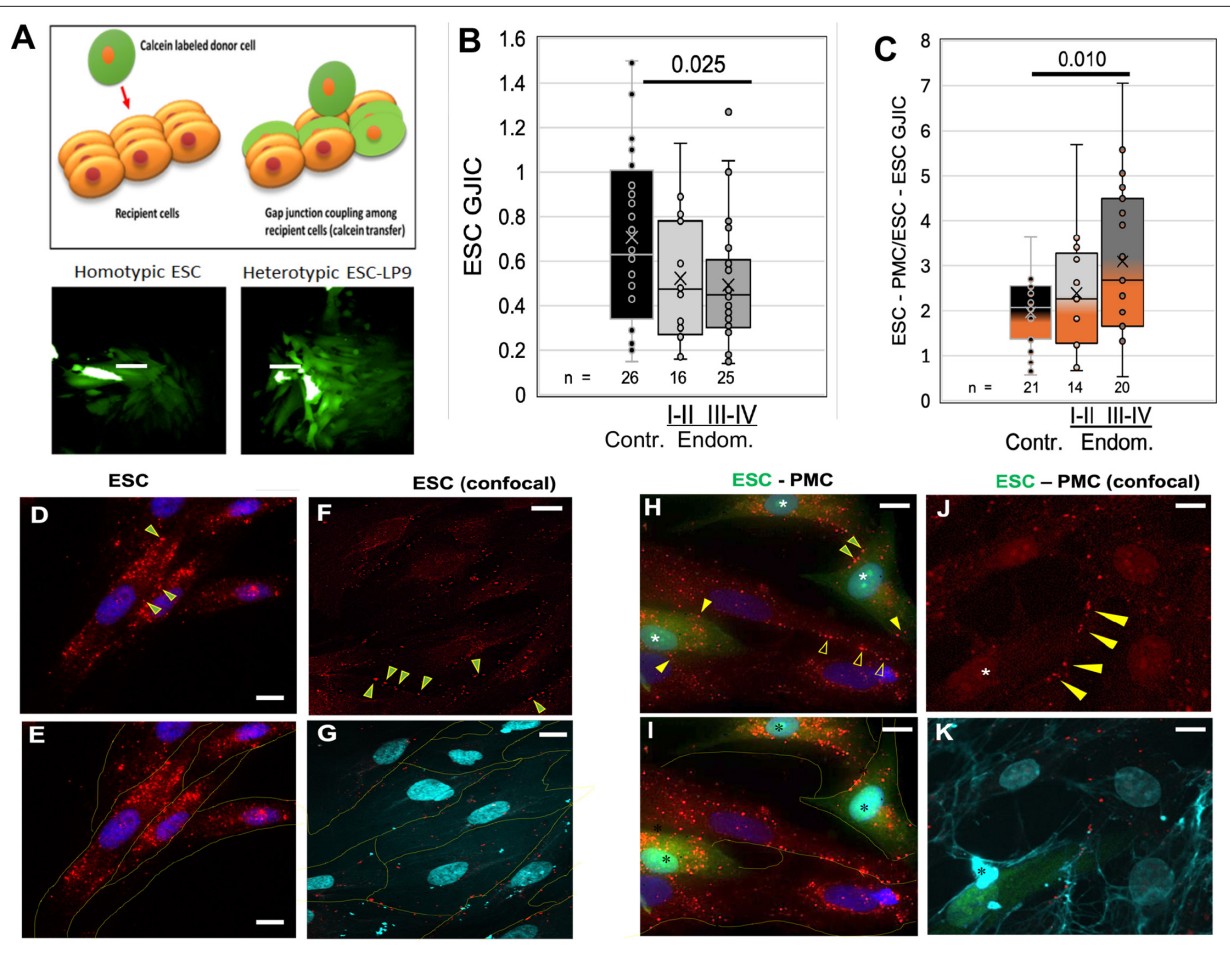

**Figure 3.** Coupling between endometrial stromal cells (ESCs) and peritoneal mesothelial cells (PMCs) is induced in endometriosis. (**A**) Gap junction intercellular coupling (GJIC) was measured by a modified 'parachute assay' where calcein-loaded donors are dropped onto a monolayer of acceptors of either the same (homocellular) or different (heterocellular) cell type, and calcein transfer is measured as a linear increase in fluorescent acceptor/donor ratio over time. Scale bars are 50 µM. (**B**) GJIC between eutopic ESCs decreased progressively with the disease, reaching significance in Endometriosis III-IV patients. (**C**) Heterocellular ESC-PMC GJIC was induced compared to ESC homocellular coupling and this increased with disease progression to 2–4.5-fold in Endometriosis III-IV patients. (**D–K**) Immunocytochemical staining of Cx43 (red), with cell outlines from phase (yellow) or membrite labeling (blue) superimposed in the lower panels. ESCs alone showed some labeling between cells (arrowheads), but most Cx43 was in intracellular pools (**D–G**). By contrast, in mixed cultures of PMCs with ESCs [labeled with cell tracker green (*)] there is less intracellular Cx43 labeing, and punctate staining of GJs between cells is increased in frequency [Arrowheads: ESC-ESC (green in yellow); PMC-PMC (hollow yellow); ESC-PMC (solid yellow)] (**H–K**). Nuclei are stained with DAPI. Scale bars are 10 µm. Number of repeats for each condition are shown in B and C, with 8–10 patients in each group. Significance based on two-tailed t-tests. Full data in *Figure 3—source data 1*.

The online version of this article includes the following source data for figure 3:

**Source data 1.** Original data tables and statistical analyses for *Figure 3B and C*.

## GJIC is required for the invasion of ESCs across a peritoneal mesothelium

As GJIC has been indirectly linked to the analogous process of extravasation (*Ito et al., 2000*: *Naoi et al., 2007*), we directly tested if GJIC between ESC and PMCs is required for invasion across the mesothelium. We employed several complementary strategies to selectively block GJIC, targeting Cx43, as it is expressed at 10 times higher levels than other connexins in both control and endometriosis ESCs as well as PMCs (*Chen et al., 2021*).

Firstly, we pre-treated both PMCs and ESCs in our invasion chambers with GAP27, a peptidomimetic of part of the second extracellular domain of Cx43 that has been shown to block the formation of gap junctions between newly contacting cells (*Evans and Leybaert, 2007*). GJIC was blocked ~85% in both controls (black) and endometriosis (gray) derived ESCs. This resulted in a reduction of invasiveness

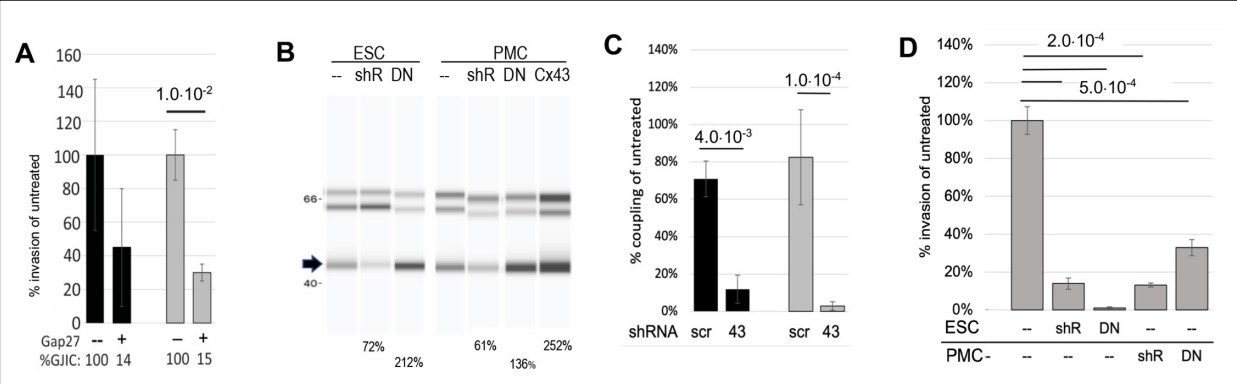

**Figure 4.** Invasiveness of endometrial stromal cells (ESCs) is dependent on Cx43 gap junction intercellular coupling (GJIC). (**A**) GAP27 peptide: Averaging ESCs from control (black bars, n=3) and endometriosis patients (gray bars, n=6), the invasion was inhibited by a peptide inhibitor of GJ channels, GAP27 (percent Gap Junction Intercellular Coupling (GJIC) compared to untreated shown below each bar). (**B-D**) shRNA: (**B**) Infection of a doxycycline-inducible Cx43 shRNA into endometriosis ESCs or LP9 peritoneal mesothelial cells (PMCs) reduced levels of Cx43 protein (arrow) compared to Laminin controls (doublet at ~66 kD), while expression of DN or wt Cx43 increased Cx43 expression levels. (% of untreated shown below gel) (see *Figure 4—source data 1* and *Figure 4—source data 2*). (**C**) Cx43 shRNA inhibited GJIC by >90% compared to scrambled shRNA in infected LP9 PMCs (black bars, n=3) and Endometriosis ESCs (Gray bars, n=4). (**D**) Invasiveness was inhibited by ~85% in Cx43 shRNA infected compared to uninfected neighbors, whether expressed in ESCs (n=7), or PMCs (n=2). DN Cx43 inhibited invasiveness by 98% when expressed in ESCs, and 65% when expressed in PMCs, where ~70% of the monolayer was infected. N represents independent tests with different shRNAs, with 10 technical replicates of each. Significance based on two-tailed t-tests. Full data in *Figure 4—source data 3*.

The online version of this article includes the following source data for figure 4:

**Source data 1.** Original full unannotated image of SDS gel in *Figure 4B*.

**Source data 2.** Annotated image of gel in *Figure 4B* with labels and bands highlighted.

**Source data 3.** Original data table and statistical analysis for *Figure 4C and D*.

of ESCs through the PMC monolayer, although this was only significant (70%) in cells derived from endometriosis patients (*Figure 4A*, n=6). In subsequent experiments, we found the effectiveness of GAP27 to block GJs and invasion to be variable, presumably due to variability between vendors and batches of the peptide. So we moved to Cx43 KD approaches.

Our first approach was to use transient transfection of siRNAs to Cx43. While these did achieve 60–70% KD of GJIC, and a significant (~40%) block of invasion, results were highly variable due to compromised cell health following transfection. This affected both the effective formation of a mesothelial barrier by as PMCs, and the motility and invasiveness of ESCs. Thus, to avoid these complicating effects of transient transfection, we moved to stable Lentiviral infection to generate PMCs and ESCs that express inducible shRNAs targeted to Cx43 (or scrambled shRNAs as control). An RFP reporter allowed us to track which cells expressed the shRNA, which averaged 61 ± 8% (n=8) of the cell population in the presence of doxycycline. Suppression of Cx43 protein levels in the total cell population was evident (*Figure 4B*) and GJIC was inhibited by 95 ± 4% (n=7) in infected ESCs (*Figure 4C*). Invasion by infected ESCs expressing Cx43 shRNA (identified by RFP expression) was reduced by 90–95% compared to uninfected cells in the same sample (*Figure 4D*). In the inverse experiment where Cx43shRNA was expressed in PMCs, the monolayer consisted of both infected (~70%) and uninfected cells, but invasion was still inhibited by ~85% (*Figure 4D*).

Finally, to ensure the inhibition of invasiveness was due to the block of GJIC, and not loss of the adhesive roles of gap junctions (since we had reduced total Cx43 levels), we used the same Lentivirus system to express a dominant negative Cx43 construct, Cx43T154A (DN Cx43) in either ESCs or PMCs, with ~70% efficiency. This DN construct forms structurally normal gap junctional plaques but prevents channel opening when co-expressed wtCx43 (*Beahm et al., 2006*). Expression of DN Cx43 increased total Cx43 levels by ~ twofold in ESCs and 1.4-fold in PMCs (*Figure 4B*). Invasive behavior of infected ESCs (identified by GFP reporter expression) was reduced by >98% compared to uninfected cells in the same sample (*Figure 4D*). Expression of the DN isoform in PMCs produced a mixed monolayer of infected (~70%) and uninfected cells and resulted in a 65% block of invasion.

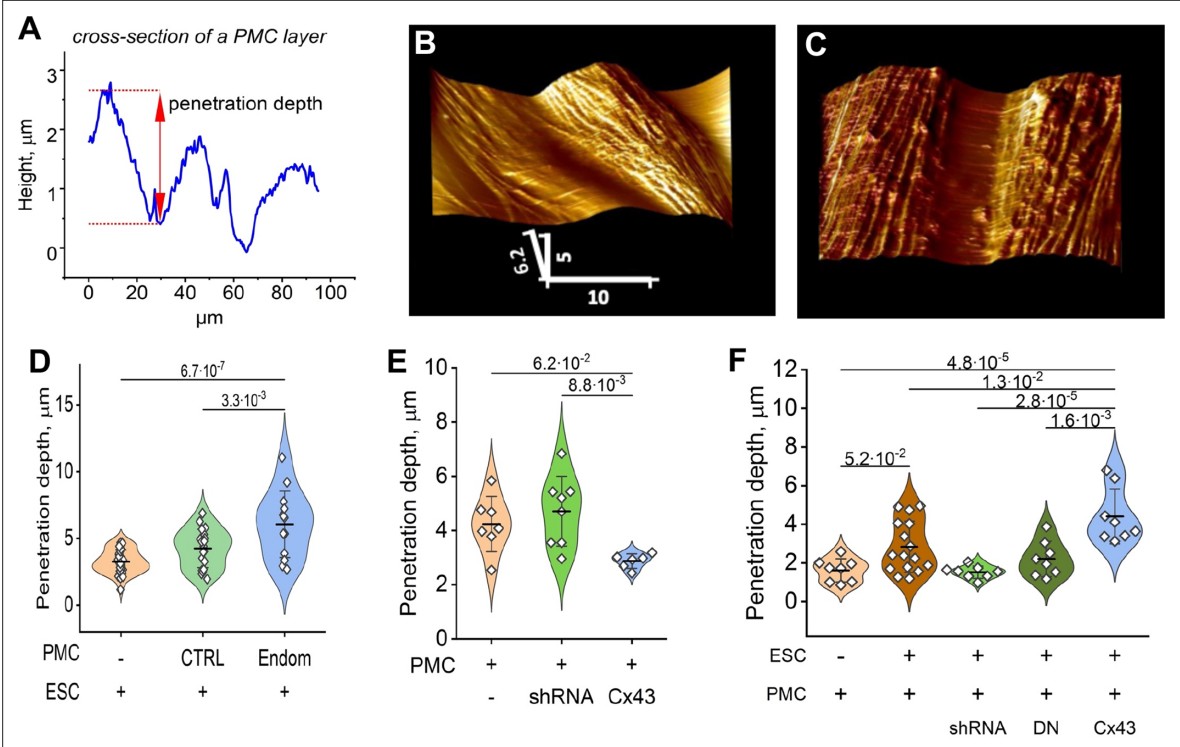

**Figure 5.** Endometrial stromal cells (ESCs) induce GJ-dependent disruption of the barrier function of a peritoneal mesothelial cell (PMC) monolayer. (**A**) Probing the topological surface of a PMC monolayer using an AFM probe under constant force allows the identification of sites of intercellular contact (where penetration of the probe is maximal). (**B–C**) 3-D reconstructions of the surface of an LP9 PMC monolayer alone (**B**) or in the presence of ESCs which induce the opening of wide gaps (**C**) (Scale bars in μm). (**D**) Penetration depth between PMCs increased more with ESCs from endometriosis than control patients. (**E**) PMC monolayer integrity (i.e. lower penetrance) is reduced by Cx43 shRNA KD and enhanced by Cx43 overexpression. (**F**) In contrast, when ESCs are dropped onto a PMC monolayer, the increased penetrance that is induced is eliminated by the expression of Cx43shRNA or DNCx43 in the PMCs and is enhanced by Cx43 overexpression. Each dot in D-F represents a single image analysis. Significance based on two-tailed t-test. Full data in *Figure 5—source data 1*, *Figure 5—source data 2* and *Figure 5—source data 3*.

The online version of this article includes the following source data for figure 5:

**Source data 1.** Original data table and statistical analysis table for *Figure 5D*.

**Source data 2.** Original data table and statistical analysis table for *Figure 5E*.

**Source data 3.** Original data table and statistical analysis table for *Figure 5E*.

While each method for blocking gap junctions may have limitations, we demonstrate that four independent approaches that block either functional GJIC between ESCs and PMCs (GAP27 or DN CX43) or expression of Cx43 in either of the cell types (si- and shRNA) all significantly reduce invasive behavior of ESCs. The fact that block in both ESCs and PMCs caused similar effects strongly implicates a role for gap junctions between these two cell types, as if hemichannels are involved, they would have to have similar effects in both cell types.

## Cx43 expression is required for both the integrity of a mesothelial barrier and its disruption by ESCs

To probe the influence of ESCs, in the presence or absence of Cx43, on the 'barrier function' of the mesothelium (ie. the intercellular contacts between PMCs comprised of tight and adhesive junctions that prevent transmigration of cells), we utilized the unique ability of AFM to probe the surface topology of a cell monolayer in real-time. ESCs from control subjects or endometriosis patients were first labeled with the membrane dye DiO, and dropped onto a PMC monolayer at a ratio of ~1:20 (ESC:PMC). After ~3 hr, the monolayer was imaged with AFM using a 'sharp' conical probe at a constant applied pressure of 1 nN to obtain a 3-D contour map of the monolayer (*Figure 5B*). This readily identified the interfaces between cells and measured the depth of penetration of the probe

between cells as a physical measure of mesothelial 'barrier function' (*Figure 5A*). ESCs from several patients all induced a widening in the gap between PMCs (*Figure 5C*), corresponding to an ~ twofold increase in penetrance, measured ~10 μm (1–2 cell diameters) from an identified dropped cell. This increase in penetration of the monolayer was seen with all ESCs but was which was more notable in ESCs from endometriosis patients (*Figure 5D*), consistent with their greater invasive potential (*Figure 1E*).

We then used Cx43 shRNA, DNCx43, and wtCx43 infected LP9-PMCs, characterized in *Figure 4B–D*, to test the dependence of these changes on GJIC. First, we observed that the 'barrier function' of a PMC monolayer in the absence of ESCs was dependent on Cx43 expression, as the degree of penetrance was reduced when Cx43 was overexpressed and increased when Cx43 was inhibited by shRNA (*Figure 5E*). This is consistent with GJs being part of the intercellular nexus, including tight and adhesions junctions, that connect cells. However, this effect was strikingly inverted when we introduced ESCs (*Figure 5F*). Cx43 overexpression in PMCs significantly enhanced penetrance in response to ESCs, while Cx43 inhibition by shRNA eliminated the effect of ESCs, so that penetrance was indistinguishable from control PMC monolayers. The only difference between the studies in *Figure 5E and F* is the presence of ESCs in the latter, implicating the formation of GJs between ESCs and PMCs as the factor that must trigger the breakdown of the PMC barrier. This directly demonstrated that this was

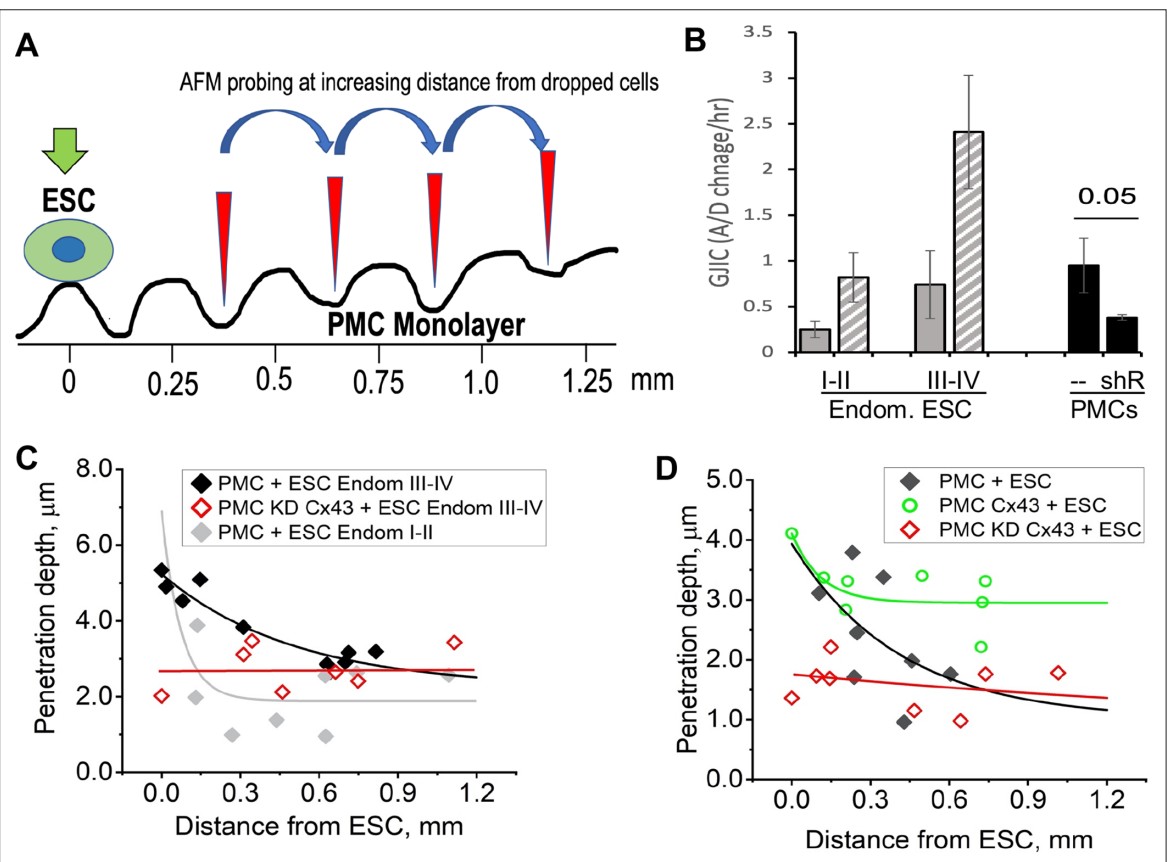

**Figure 6.** Disruption of the mesothelial barrier by endometrial stromal cells (ESCs) is propagated through mesothelial gap junctions. (**A**) Using a constant force of 1 nN, the AFM tip was moved over the peritoneal mesothelial cell (PMC) monolayer progressively further away from a dropped DiI labeled ESC. (**B**) ESCs from an endometriosis III-IV patient showed greater homocellular (solid gray) and induced heterocellular gap junction intercellular coupling (GJIC) with PMCs (striped gray) than those from an endometriosis I-II patient. GJIC of LP9 PMCs was also measured and shown to be decreased by 60% through an expression of Cx43shRNA (black). (**C**) Penetration through the LP9 PMC monolayer decayed with distance from the dropped ESC much faster in the poorly coupled Endo I-II ESCs (gray) than the better coupled Endo III-IV ESCs (black). Penetration of the monolayer was eliminated by KD of Cx43 in PMCs (red). (**D**) Conversely, the decay in penetration of the PMC monolayer induced by Endo III-IV ESC cells (black) was greatly reduced by over-expression of Cx43 in PMCs (green). Full data in *Figure 6—source data 1*.

The online version of this article includes the following source data for figure 6:

**Source data 1.** Original data sets for *Figure 6C and D*.

dependent on Cx43 channels, as expression of a DN Cx43 in PMCs, which suppresses coupling but maintains the gap junction structures, and all their adhesive and structural properties, also prevented barrier breakdown similarly to Cx43 shRNA.

This raised the question that if GJs pass signals from ESCs to PMCs that promote the breakdown of the mesothelial barrier, do gap junctions between PMCs also play a role in propagating such a signal distribution through the mesothelium. To test this, we dropped ESCs at a lower density (1:50 ratio with PMCs) and measured the degree of penetrance between PMCs at increasing distances from a single contacting ESC (*Figure 6A*). ESCs from stage I-II patients, which showed only modest induction of ESC-PMC coupling, were compared with ESCs from stage III-IV patients, which showed much greater levels of ESC-PMC coupling (*Figure 6B*). When penetrance was plotted against distance from a dropped ESC, the influence of the ESCs decayed to background levels at much greater rates in the case of poorly coupled ESCs (Endom. stage I-II) than well-coupled ones (Endom. stage III-IV), propagating over distances of 200 μm to >700 μm, respectively (*Figure 6C*). As expected, knockdown of Cx43 by shRNA in the LP9 cells eliminated all effects of the ESCs. However, overexpression of Cx43 in LP9 PMCs caused a dramatic extension of the propagation range to beyond the limits of our recording at 800 μM (*Figure 6D*), corresponding to over 40 cells from the dropped ESC.

## Discussion

Despite afflicting 10% of the female population, the etiology of endometriosis is still the subject of debate. Retrograde menstruation of endometrium into the peritoneal cavity is the most widely accepted theory to explain most endometriosis cases. However, why is endometriosis seen in only 10% of women, when ~90% display retrograde menstruation? Are there specific changes in the uterine endometrium (the 'seed') or the peritoneal lining (the 'soil') that predispose patients to develop the disease? Most studies have focused on expression changes in endometrial cells in utero, from established lesions, or on the inflammatory sequalae of the disease. Few studies have examined the initial stages of lesion formation that could provide mechanistic insights into disease pathophysiology and most directly address the issue of a 'seed' or 'soil' origin. We have taken a reductionist approach to the problem by isolating and characterizing each major cell type involved in initial lesion formation from control (19) and endometriosis (22) patients: endometrial epithelial (glandular) and stromal (supporting) cells (EECs and ESCs, respectively) from control and endometriosis uterine biopsies, and PMCs, both established LP9 cells and isolated from control and endometriosis patients (characterized in *Acosta Go et al., 2023*).

Of the endometrial cells, the major focus was on ESCs, as they proved more adhesive to PMCs, more motile, and ultimately twice as invasive as EECs from the same patients. ESCs from endometriosis compared to control patients were also more adhesive to PMCs, more motile (~ twofold), and much more invasive across PMCs (2–6 fold depending on disease stage). EECs were mixed with ESCs to mimic in vivo conditions, invasiveness was further enhanced, but this was only significant in endometriosis samples, which also showed a higher fraction of EECs in the invading cell population than controls (40% compared to 15%). Our experiments were conducted on primary cells cultured from pipelle endometrial biopsies, which leaves open the possibility that stem cells present in the endometrium could be included in our sample, although they may differentiate into one of the main cell types in the culture.

Our observations that endometriosis-derived ESCs show increased responsiveness to other cell types, including enhanced motility and adhesion to, gap junction communication with, and invasion across PMCs, combined with prior findings that endometriosis endometrial cells show enhanced repression of apoptosis (*Taniguchi et al., 2011*), and immune avoidance (*Han et al., 2015*; *Björk et al., 2024*), strongly implicate changes in the endometrium as causative of the disease. The most direct deduction from these results is that the 10% of patients who develop endometriosis are distinguished from the 80% that have retrograde menstruation without endometriosis by pre-existing changes in the endometrial lining that predispose the cells to an invasive phenotype.

One important question is the extent to which endometrial cell behaviors are affected by the menstrual phase, birth control, or other variables between patients. We compared invasiveness, motility, and induction of GJIC by PMCs between control and endometriosis ESC in patients from the proliferative or secretory phases of the menstrual cycle or those on oral contraceptives. While the limited numbers precluded applying robust statistics, under all menstrual conditions the endometriosis

cells showed enhanced activity of each phenotype. Among endometriosis patients, where we had 3–6 patients in each group, we also compared each ESC phenotype from patients in different menstrual conditions (cycle stage or oral contraceptives). No statistically significant differences were observed, suggesting modest hormonal influences on the aspects of ESC behavior associated with invasion. Any minor differences are outweighed by the disease phenotype.

Since we could not assess endometrial samples patients prior to disease manifestation, it is certainly possible that some of the changes we observe may be a consequence of the disease through feedback from lesions in the peritoneum that can globally affect hormonal levels and inflammatory responses. Indeed, such effects could explain the observations in *Nirgianakis et al., 2020* that a significant number of patients (48%) presenting initially with superficial lesions can show more deep infiltrating lesions on recurrence. Delineating the degree to which endometrial changes pre-exist disease onset will remain a challenge based on both practical and ethical considerations governing human trials. The only thing that is clear from the studies of *Acosta Go et al., 2023* is that the invasive behavior of ESCs is not influenced by PMC origin (from control or endometriosis patients), only by ESC origin.

While others have compared other aspects of endometrial behavior (*Taniguchi et al., 2011*; *Han et al., 2015*; *Björk et al., 2024*), we have focused on their invasive behavior and interactions at the mesothelial lining. The enhanced invasiveness of eutopic ESCs from Endometriosis patients across PMCs seems in large part to be due to their enhanced responsiveness to PMC signals that increase GJIC and motility. This result is consistent with our prior CyTOF single-cell Mass Spectroscopy comparisons of ESCs alone and in co-culture with PMCs that showed much larger shifts in expression of markers of EMT plasticity (ZEB1, SNAIL1, TWIST) in endometriosis than control-derived ESCs (*Lin et al., 2020*). That ESCs also increase invasiveness when co-cultured with their cognate EECs, suggests that the hypersensitivity of endometriosis ESCs is not restricted to PMCs.

Many of these changes parallel those observed in metastatic cancer, including the enhanced motility, target-induced changes in EMT plasticity (*Lambert et al., 2017*), and the ability to breakdown tissue barriers during intra- and extravasation and metastatic invasion (*Reymond et al., 2013*). In the latter processes, one of the earliest steps is the formation of gap junctions between the tumor cell and endothelium (*el-Sabban and Pauli, 1991* and *el-Sabban and Pauli, 1994*; *Ito et al., 2000*; *Naoi et al., 2007*) or target tissue (*Stoletov et al., 2013*; *Hong et al., 2015*; *Chen et al., 2016b*). In fact, GJ are typically suppressed in primary tumors, but need to reactivate in order to metastasize (*Wu and Wang, 2019*), which is often associated with more efficient transport of connexins to the cell surface (*Kanczuga-Koda et al., 2006*). We demonstrate here a very analogous process in endometriosis. GJ expression (*Yu et al., 2014*; *Chen et al., 2021*) and cell coupling are suppressed ectopically but then GJ coupling is strongly induced ectopically when ESCs encounter PMCs through the trafficking of intracellular Cx43 stores to cell-cell interfaces. By further analogy with metastasis, the invasiveness of ESCs across the mesothelium is shown here to be highly correlated with their invasiveness across an endothelium. This is important in terms of disease, as the low incidence of endometriosis outside of the peritoneum has been used to argue that lesions may arise from non-endometrial cells such as stem cells or Mullerian remnants. However, it seems that the same features that make ESCs more invasive in the peritoneum In endometriosis, also increase their chances of entering the bloodstream.

GJ coupling has been associated with enhanced motility in cancer cells (*Zhang et al., 2015*; *Polusani et al., 2016*) and may play a role in PMC induction of ESC motility seen here, although this was not directly tested. However, we do clearly demonstrate that this induced GJ coupling between ESCs and PMCs is required for disruption of the mesothelial barrier and invasion, something that has not been definitively established in metastasis. As illustrated in *Figure 7*, we propose that contact between ESCs and PMCs induces the trafficking of Cx43 to the surface and formation of ESC-PMC gap junctions that mediate the transfer of yet-to-be-identified intercellular signals that initiate the disruption of the intercellular junctional nexus that prevents trans-mesothelial migration. The adhesive roles of gap junctions do not appear to play a major role, as the DN Cx43T154A, which preserves junctional structures, but prevents channel opening (*Beahm et al., 2006*), also inhibits invasion. Release of signals through Cx43 hemichannels also seems unlikely, since invasion can be similarly prevented by KD of Cx43 in either ESCs or PMCs.

It is interesting to note that we demonstrate here that PMC GJs promote mesothelial integrity under normal conditions, possibly through nucleating other junctional structures between cells (e.g. tight and adhesions junctions) with whom they share many accessory and cytoskeletal binding proteins

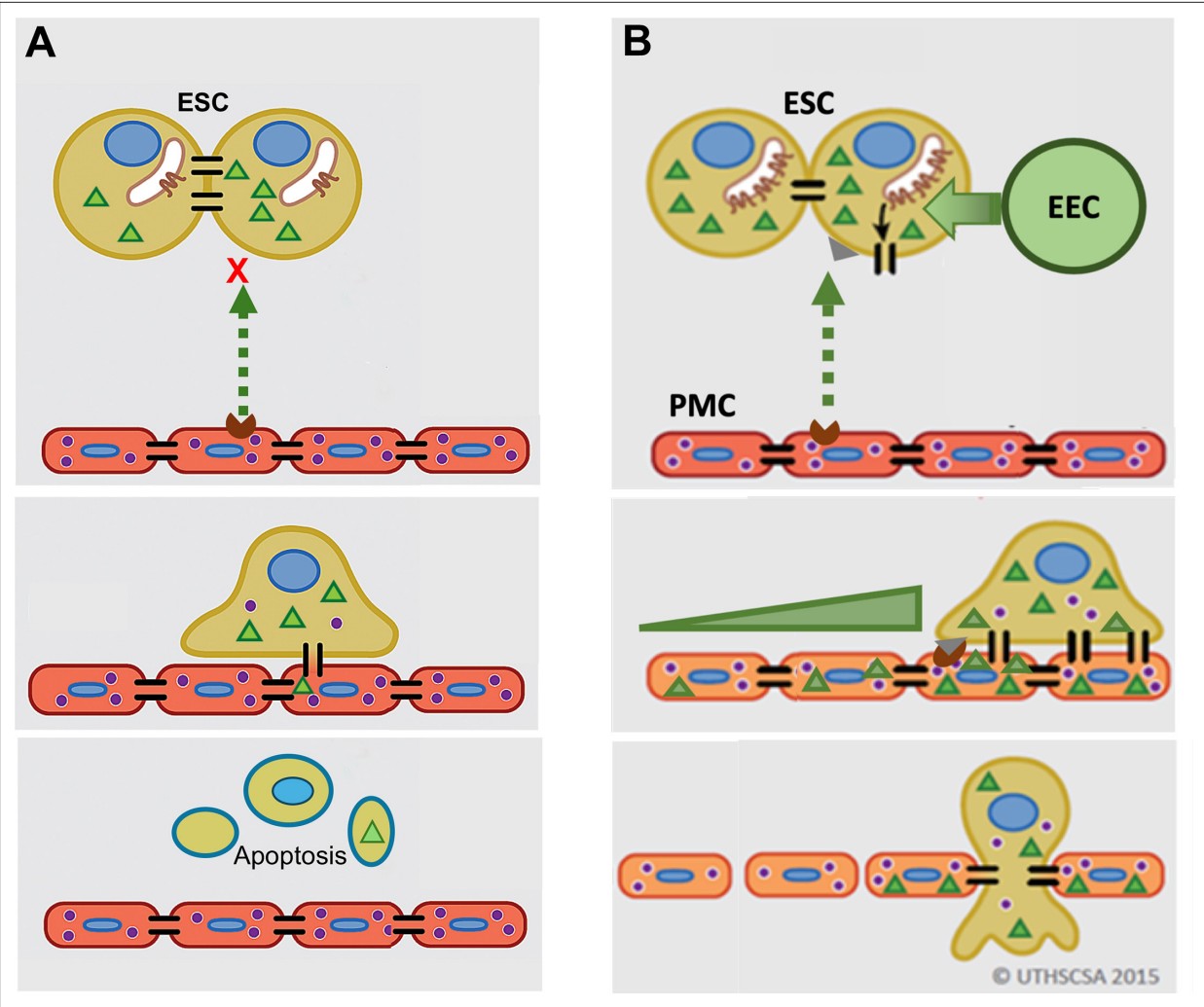

**Figure 7.** Model of gap junction intercellular coupling (GJIC) induction of trans-mesothelial invasion. (**A**) In healthy patients, when endometrial cells (light brown) encounter a mesothelium (brick red) following arrival in the peritoneum via retrograde menstruation, there is limited GJIC between endometrial stromal cells (ESCs) and peritoneal mesothelial cells (PMCs). ESCs also likely undergo apoptosis. (**B**) In endometriosis, interactions with mesothelial cells trigger Cx43 trafficking to the cell surface and a significant enhancement of GJIC. The increased GJIC mediates the transfer of signals to PMCs (green triangles), which propagate through the mesothelium, inducing disruption of the adhesive and tight junctions between PMCs, facilitating the invasion of the ESCs. There would also be a passage of signals from PMCs to ESCs (purple dots) that induce further changes in ESCs that could promote invasion (*Lin et al., 2022*). EECs (green cells) show minimal invasion alone, but can enhance ESC invasion, and in endometriosis invade with ESCs.

(e.g. ZO1, β-catenin, etc.). But apparently, this proves to be an Achilles heel when ESCs arrive in the peritoneum, as now the GJs between PMCs mediate further propagation of intercellular signals leading to a breakdown of mesothelial intercellular contacts at significant distances from the site of ESC contact (*Figure 7B*, **lower panel**).

Together, these studies demonstrate that eutopic ESCs (i.e. from the uterine endometrium) are very distinct in endometriosis and control patients, the former being characterized by enhanced responsiveness to interactions with EECs and PMCs that promote invasive behavior. Gap junctions between ESCs and PMCs, and within the mesothelium, are shown to be critical in initiating lesion formation through mutually induced changes in the phenotypes of both cells. Together these results demonstrate that changes within the uterine endometrium prime ESCs to be invasive once they reach the peritoneal cavity. There seem to be many parallels with the development of metastatic potential in cancer cells, which is likely determined before they leave the primary tumor, and also involves an induction of gap junction formation that facilitates invasive behavior. Our data also supports a 'seed'

(endometrium) rather than 'soil' (mesothelium), origin for endometriosis, which is more definitively established by *Acosta Go et al., 2023*.

## Materials and methods

### Primary endometrial epithelial cell isolation from endometrial biopsies

Primary ESCs and EECs were isolated from endometrial biopsies obtained from women with and without endometriosis under IRB protocol # 20070728 HR (8-31-23). All women provided informed consent prior to participating in this Institutional Review Board-approved protocol. Study subjects were premenopausal women between 30 and 45 y of age with regular menstrual cycles, or in some cases on Oral Contraceptives, undergoing laparoscopic surgery for gynecologic indications (*Table 1*). Women with pelvic inflammatory disease/hydrosalpinx, endometrial polyps, or submucosal fibroids were excluded. Two control patients (H11 and H20) and one Endometriosis patient (31) were found to have ovarian cysts at the time of surgery. Endometriosis was staged according to the revised American Society for Reproductive Medicine (ASRM) criteria and confirmed by histopathologic review of peritoneal or cyst wall biopsy in all cases. Fertile women undergoing tubal sterilization and without endometriosis at surgery were considered healthy controls. Menstrual cycle phase (proliferative or secretory) was determined by cycle history and confirmed by serum estradiol and progesterone levels when available. Endometrial tissue was obtained by pipelle biopsy at the time of laparoscopic surgery. In some patients, during laparoscopy for definitive diagnosis of disease, small biopsies of the peritoneum were also taken both in the vicinity of, and distant to, identified lesions. These samples were kept on ice for <2 hr before embedding in OCT and freezing and storage at –80 °C for subsequent immunocytochemistry.

Pipelle endometrial biopsy material was dissociated by shaking in 5 mg/ml collagenase and 2.5 mg/ml DNase in Hanks Balanced Salt Solution at 37 °C for 1 hr. Isolation of primary ESCs and EECs from the biopsies was performed using a combination of straining (45 uM nylon filter) and differential sedimentation (EECs cluster and sediment faster), followed by differential attachment (EECs adhere less well to culture plates), in a modification of the method developed by *Kirk and Irwin, 1980* used in prior studies (*De La Garza et al., 2012*; *Chen et al., 2016b*). In some experiments, the differential attachment step was replaced by using an Ep-CAM affinity column to enrich EECs. Both methods achieve about 97% purity for EECs and ESCs, as illustrated in *Figure 1—figure supplement 1* by immunostaining for epithelial [(EpCAM- ab71916 from Abcam, Waltham, MA) and CK 7 (ab902 and 1598 from Abcam)] and stromal [Vimentin (MA1-10459 from Thermo Fisher, Waltham, MA; NBP1-92687 from NovusBio, Centennial, CO)] markers.

### Cell culture

Primary ESCs were cultured in Dulbecco's Modified Eagle Medium (DMEM)/F12 (1:1) (Gibco, Buffalo, NY) containing antibiotic/antimycotic mix (Gibco, Buffalo, NY), 10 µg/ml insulin (Sigma, St. Louis, MO) and 10% heat-inactivated fetal bovine serum (FBS - Gibco, Buffalo, NY) as described previously (*Ferreira et al., 2008*). EECs were cultured in MCDB/Medium 199/MEMα (1:1:0.6) containing antibiotic/antimycotic mix, 10 ug/ml insulin, D-Glucose (0.45%) (Sigma, St. Louis, MO), Gluta-Max and 10% FBS (Gibco). Prolonged culture was in defined KSFM with supplement, 1% FCS, and antibiotics/antimycotics (Gibco) to preserve the differentiated state of the EECs (*Chen et al., 2016b*) although this generally was only possible to 3–4 passages. All experiments were performed using low passages (≤4) to avoid loss of differentiated characteristics. Established LP9 cells (Karyotype verified from NIA Aging Cell Culture Repository #AG07086 PDL 4.84 passage 6, Coriell Institute, Camden, NJ) were used as a model for peritoneal mesothelium and cultured as described previously (*De La Garza et al., 2012*; *Liu et al., 2009*) and grown in MCDB 131.Medium 199 (1:1 - Gibco) with 15% FBS, sodium pyruvate, Gluta-Max, antibiotic/antimycotic mix (Gibco), 20 ng/ml hEGF, and 0.4 ng/ml hydrocortisone (Sigma, St. Louis, MO). All cells used were confirmed to be mycoplasma-free. Previous studies, including our work, have validated and used LP9 cells as a model peritoneal mesothelial line for peritoneal invasion by endometrial cells (*Nair et al., 2008*). Primary peritoneal mesothelial cells from control or endometriosis patients (from regions not containing lesions) were cultured form explants as described in *Acosta Go et al., 2023*. Identity and purity of all cell cultures were confirmed by

immunocytochemistry, using antibodies for Vimentin or CD10 for ESCs, CK 7 for EECs (*Figure 1—figure supplement 1*), and Calretinin (ab92341- Abcam) for PMCs.

## Trans-mesothelial invasion assay

The 3-D invasion assay modeling trans-mesothelial invasion (*Figure 1C*) has been described previously (*De La Garza et al., 2012*; *Ferreira et al., 2008*; *Nair et al., 2008*). Briefly, LP9 PMCs were grown to confluence in 24-well invasion chamber inserts containing growth-factor-reduced Matrigel, coated on 8 µm pore membranes (Corning, NY). ESCs were labeled with the lipophilic dyes DiO (Invitrogen/Thermo Fisher), trypsinized, and counted, prior to dropping onto the confluent layer of LP9 PMCs in the prepared inserts (~20,000 cells per insert). Media above the insert was replaced immediately prior to the assay with serum-free stromal media, and below with 1% serum in stromal media, although other serum gradients were tested After 24 hr incubation, non-invading cells on the upper surface of the insert were mechanically removed. Invading cells on the bottom of the membrane insert, were stained with DAPI, and 10 fields were counted using an Inverted Nikon 2000 fluorescence microscope with a 20 x objective, confirming in each case that the DAPI-stained nuclei were associated with DiO staining. In ESC/EEC mixed cell studies, the cells were labeled before mixing with DiO and DiI, respectively, and serum-free stromal media was used on the top with 1% FBS containing stromal media on the bottom. In the case of mixed stromal and epithelial invasion studies, LP9 mesothelial media was used on top and stromal media was used on the bottom (i.e. no attractive serum gradient). Invasion assays for each cell type were performed in triplicate.

## Block of gap junction coupling

To test the role of gap junctions in the invasive process, we initially pretreated both the monolayer and dropped cells for 24 hr with 300 uM GAP27 (Zealand Pharma, Copenhagen, Denmark) (*Figure 4A*). In other experiments (data not shown), Cx43 KD was achieved by a 24- hr pre-treatment of the LP9 monolayer with a combination of two siRNAs to Cx43 (10 pmoles/well or 5 nM final concentration) - Ambion Silencer Select in OptiMEM (Gibco, NY) with RNAiMAX (1/100 dilution, Invitrogen/Thermo Fisher), diluted 1:1 with assay media, per manufacturer's instructions. In a final set of experiments (*Figure 4B–D*), ESCs, or PMCs were infected with Lentiviruses expressing one of four doxycycline-inducible shRNAs directed to Cx43, along with a pIRES RFP to identify the cells expressing the shRNA (TRIPZ vectors – Dharmacon, Lafayette, CO). Lentiviral vectors constructed in-house expressing wt or DN Cx43(T154A) with a bicistronic GFP reporter were also used in some experiments.

## Western blotting

To assess the effectiveness of viral infections with shRNA to Cx43 or expression of wt or DN versions of Cx43, ~107 cells were lysed in 1 ml of standard RIPA buffer, insoluble material spun out at 12,000 rpm for 10 min prior to assessing protein concentration by a BCA assay kit (#23225-Thermo Fisher). 1.2 ug of protein per sample is then solubilized in standard SDS loading buffer with 1 mM DTT for 30 min at RT, then loaded on an automated Western System (Biotechne, Minneapolis, MN) using a 12-230kD Wes separation module cassette according to the manufacturer's instructions. The individual capillary gels within each cassette allow for band fixation, antibody labeling, and visualization (using a fluorescent master mix) within the gel. A biotinylated marker set of proteins was run in one lane. Antibodies used were anti-Laminin A/C (#2032 Cell Signaling Technology Danvers, MA) and anti-Cx43 (#3512 Cell Signaling Technology), both at 1/50 dilution.

## Immunocytochemistry

ESCs or EECs are plated at ~50 K cells per well onto eight chamber slides (Nunc LabTech II, ….) pre-coated with 100 ug/ml poly-D- Lysine for 30 min at 37 °C and grown to 50–90% confluence. In co-culture experiments of ESCs with LP9 PMCs, LP9 cells were plated first and grown overnight to 70–90% confluence before dropping ~20 K ESCs pre-labeled with 1/1000 dilution of CellTracker Green (#C2925, Invitrogen/Thermo Fisher) in serum-free media for 30 min at 37 °C. After 4 hr to allow ESCs to attach, cells were either fixed, or in some cases pre-treated and stained with Membrite Fix dye (#30,092 T, Biotium, Freemont, CA) at 1/1000 dilution per the manufacturer's instructions to visualize membranes. All cells were washed with PBS with 1 mM Ca$^{++}$/Mg$^{++}$ (CaPBS) prior to fixation with 2% paraformaldehyde (Sigma, St. Louis, MO) for 15 min at RT. Further CaPBS washed preceded

permeabilization with 0.25% triton X-100 and 1% glycine in PBS (15 min at RT) and subsequent blocking of non-specific binding with1% Bovine Serum Albumin (BSA) (Sigma, St. Louis, MO) in 0.5% Tween-20 (1 hr at 37 °C or 4 °C overnight). Primary antibody staining was for 3 hr at RT or overnight at 4 °C in 0.1%BSA, 0.2% Tween-20 in PBS. Primary antibodies used were: anti-Cx43 (#3512 Cell Signaling Technology) at 1:100 dilution; anti-Cytokeratin 7 for EECs (#902-Abcam) at 1/1000, and anti-Vimentin (#1–92687, NovusBio) at 1/5000. After CaPBS washes, secondary antibodies to the appropriate species conjugated to Alexa 488 or Alexa 594 (#s 10680 and 11037-Invitrogen) were used at 1:1000 concentration for 1 hr. at room temperature in the dark. After final washes in CaPBS, the chamber grid is removed and a coverslip mounted with slow-fade Diamond mountant with 4',6-diamidino-2-phenylindole (DAPI) (#S36964, Invitrogen) to visualize the nuclei. Cells were imaged on a Nikon 2000 inverted epi-fluorescent microscope. In some cases, superimposed phase images were used to trace the membrane contacts between cells for clarity in visualizing Cx43 localization.

## Motility assays

Motility was assessed by a wound healing assay illustrated in *Figure 2A–B*. Cultures are grown to confluence in a 96-well plate format before being mechanically wounded and washed. Wound closure is measured every 3 hr in the Incucyte automated cell monitoring system (Essen Biosciences/Sartorius, MI) over ~3 d. Mixed cultures were plated with 2/3rd primary ESCs with 1/3rd LP9-PMCs. As we found that dye labeling of the cells can affect motility, only bulk migration of the whole culture was measured. % wound closure was platted against time and the linear portion fitted by regression analysis to provide the rates shown. All assays were performed in quadruplicate wells.

## Homo-cellular and hetero-cellular GJIC assays

GJIC was measured using a novel automated parachute assay. Recipient cells are grown to confluence in a 96 cell flat bottomed plate, and the media changed to (Phenol Red-free DMEM, sodium pyruvate, and 5%FBS – Assay Media) immediately before the assay. Donor cells in separate wells are incubated for 20 min with 10 uM calcein AM (Invitrogen/Thermo Fisher), a membrane-permeable dye that on cleavage by intracellular esterases becomes membrane impermeable, but permeable to gap junctions. After washing, trypsinization, and addition of assay media, ~2500 calcein-labeled donor cells per well are dropped ('parachuted') onto the recipient cell layer, and calcein transfer between donor and recipient cells is observed by fluorescent microscopic imaging (*Figure 3A*). For homo-cellular interactions, ESCs, EECs or LP9 donor cells were parachuted onto recipient cells of the same type. For hetero-cellular GJIC assays, ESCs or EECs were parachuted onto LP9 recipient cells. Fluorescent, bright field, and digital phase contrast images of 10–15 fields per well were captured on an Operetta automated microscope (Perkin Elmer) at 30- min intervals for approximately 2 hr. A program (developed in consultation with Perkin Elmer) allowed the identification of all cells on the plate, (from phase contrast image), original donors (5–15 per field), and dye-filled recipients (based on calcein intensity). Data are expressed as # of fluorescent recipient cells/# of donor cells for each condition (A/D ratio), plotted over time, and a linear regression line drawn through the data, with the slope used as a measure of coupling and regression coefficient (typically >0.8) used as a measure of assay reliability.

## AFM measurements of cell-cell adhesion and mesothelial integrity

We applied a Nanoscope Catalyst atomic force microscope (AFM, Bruker) interfaced with an epi-fluorescent inverted microscope Eclipse Ti (Nikon, Melville, NY). AFM images were acquired with the Peak Force Quantitative Nanomechanical Mapping (QNM) mode with cells immersed in appropriate culture media. ScanAsyst probes (Bruker, Billerica, MA) with the nominal spring constant 0.4 N/m were used for imaging. The exact spring constant for each probe was determined with the thermal noise method (*Butt and Jaschke, 1995*). For each cell culture dish at least five fields 100 by 100 μm were collected with the Peak Force set point of 2nN, and electronic resolution of 256 by 256 pixels. Nanomechanical data were processed with Nanoscope Analysis software v.1.7 (Bruker) using retrace images.

### Cell-to-cell adhesion

We attached a tester cell to a cantilever of a tipless probe MLCT-O10 (Bruker, cantilever A, spring constant 0.07 N/m) using polyethyleneimine (PEI) as a glue (*Friedrichs et al., 2013*; *Figure 1A*).

Briefly, the probes were immersed in 0.01% PEI in water for 30 min. Tester cell's attachment to a culture dish was weakened by replacement of the culture medium with a non-enzymatic cell dissociation solution (Millipore) for 15–30 min in a cell culture incubator (37 °C, 5%CO). Next, a single tester cell loosely attached to a culture dish was attached to a PEI-covered cantilever by pressing it at 1nN for 5–10 min. After visual inspection of successful cell attachment, the tester cell was lifted and transferred to a dish containing single tested cells. Then the tester cell was positioned over a tested cell and the cantilever slowly lowered till cell-cell interactions were detected with a force plot. The cells were left interacting for 30–180 sec at forces 0.5–5 nN and then the tester cell was lifted. During this step, a force plot was recorded, and the collected data was applied to calculate cell – cell adhesion parameters. The force plots were baseline corrected and a maximum of adhesion between cells during their detachment was calculated (units of force, Newton) (*Taubenberger et al., 2014*; *Dufrêne et al., 2017*).

### Integrity of LP9 mesothelial monolayer

LP9 cells were grown to confluence in a 60 mm culture dish. ESC cells grown in separate wells were stained with DiO, suspended, and dropped on to the LP9 monolayer at either a 1:50 or 1:20 ratio to the LP9 cells. In cases where cell mixes were used, ESCs and EECs were labeled with different dyes (DiI and DiO, respectively) prior to mixing in equal numbers and dropping onto PMCs. Three hours later the cells were imaged by AFM (*Figure 5B–C*). To calculate a tip penetration depth, cell boundaries were identified using images collected by the peak force error (PFE) channel. To exclude gap areas between cells or areas of cells growing in multilayers, PFE images were overlaid with height channel images after processing them with the flatten function of first order. Tip penetration was calculated based on a height histogram of all data points using a difference between the prevalent maximum of cell monolayer height and the prevalent maximum depth between cells accessible for the tip (*Figure 5A*).

## Acknowledgements

We would like to express gratitude to: the CPRIT-funded High Throughput Screening Facility which helped with the assessment of GJIC; the CPRIT-funded Bioanalytics and Single-Cell Core (BASiC) facility at the University of Texas Health San Antonio for AFM analyses; the clinical teams who helped in the collection of patient samples, particularly Jessica Perry at UTHealth SA, Peter Binkley, curator of the UTHealth SA Ob/Gyn tissue bank, Dr. Robert S Schenken who collected initial samples used in our GAP27 studies and Janan van Osdell and Somer Baburek from Hear Biotech Inc who provided patient samples for motility studies; We would also like to thank Taryn Olivas and Taylor Williams who made critical contributions to the early characterization of patient samples and Dr. Li-Ling Lin for her insightful observations on CyTOF studies that helped us interpret our functional findings.This work was initially supported by an NICHD award RO1HD109027 to BJN, the Endometriosis Foundation of America (NBK), and generous earlier internal support from UT Health San Antonio through the LSOM Women's Health Initiative (BJN and NBK), a Presidential Entrepreneurial Fund Award (NBK) and a pilot award from the Circle of Hope through the Mays Cancer Center (BJN). GJIC measurements were performed in the High Throughput Facility within the CIDD, funded by CPRIT, (# RP160844) and the Institute for Integration of Medicine and Science (NIH-NCATS grant # UL1 TR 002645). AFM measurements were performed in the BASiC facility at the University of Texas Health San Antonio, which receives funding from CPRIT grant # RP150600.

## Additional information

### Competing interests

Nameer B Kirma, Bruce J Nicholson: co-founder and shareholder of Hear Biotech Inc, developing a diagnostic for endometriosis in part based on these findings. The other authors declare that no competing interests exist.

## Funding

| Funder | Grant reference number | Author |
|---|---|---|
| Eunice Kennedy Shriver National Institute of Child Health and Human Development | R01HD109027 | Bruce J Nicholson |
| Cancer Prevention and Research Institute of Texas | RP160844 | Bruce J Nicholson |
| National Center for Advancing Translational Sciences | UL1 TR 002645 | Bruce J Nicholson |
| Cancer Prevention and Research Institute of Texas | RP150600. | Nameer B Kirma |

The funders had no role in study design, data collection and interpretation, or the decision to submit the work for publication.

## Author contributions

Chun-Wei Chen, Maria Gaczynska, Pawel Osmulski, Formal analysis, Investigation, Methodology, Writing – review and editing; Jeffery B Chavez, Ritikaa Kumar, Srikanth R Polusani, Formal analysis, Investigation, Methodology; Virginia Arlene Go, Resources, Formal analysis, Investigation, Methodology, Writing – review and editing; Ahvani Pant, Investigation; Anushka Jain, Investigation, Performed the experiments shown in Figure 1G, which were added to address an issue raised by the Reviewing Editor; Matthew J Hart, Software, Investigation, Methodology; Randal D Robinson, Conceptualization, Supervision, Methodology, Writing – review and editing; Nameer B Kirma, Conceptualization, Resources, Supervision, Funding acquisition, Methodology, Writing – review and editing; Bruce J Nicholson, Conceptualization, Resources, Data curation, Formal analysis, Supervision, Funding acquisition, Investigation, Methodology, Writing – original draft, Project administration, Writing – review and editing

## Author ORCIDs

Chun-Wei Chen (ID) https://orcid.org/0000-0002-3318-0349
Nameer B Kirma (ID) https://orcid.org/0000-0002-4657-3774
Bruce J Nicholson (ID) https://orcid.org/0000-0003-1649-7173

## Ethics

Explicit patient consent was obtained for all endometrial samples used in this study. All samples used in experiments were de-identified to the investigators. Approval for all protocols was obtained through the IRB at the Universoty of Texas Health San Antonio, IRB protocol # 20070728HR (8-31-23).

## Decision letter and Author response

Decision letter https://doi.org/10.7554/eLife.94778.sa1
Author response https://doi.org/10.7554/eLife.94778.sa2

# Additional files

## Supplementary files

MDAR checklist

## Data availability

As described in the MDAR, primary data results are reported as source data for all figures except Figure 5, where data points are shown directly on the plots.Patient data is described, but samples are not available for extrenal use based on patient consent limitations. DNA sequences used for silencing studies are commercially available, and the source listed.

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
