## [Editor Report]

The authors show how gap junction proteins are induced to be expressed at the cell surface when the endometrium interacts with the mesothelium, and that this induction is much higher in endometrial stromal cells from endometriosis patients than controls. Strengths include the use of various methods, with useful results demonstrating that gap junction coupling between endometrial stromal cells and the mesothelium is required for invasion in vitro. These findings provide solid support for the role of the endometrium in allowing endometriosis to be an invasive disorder.

---

## [Decision Letter]

**Decision letter after peer review:**

**[Editors’ note: the authors submitted for reconsideration following the decision after peer review. What follows is the decision letter after the first round of review.]**

Thank you for submitting your work entitled "Endometrial Gap Junction Expression -Early Indicators of Endometriosis and Integral to Invasiveness" for consideration by *eLife*. Your article has been reviewed by 3 peer reviewers, including Sang Jun Han as the Reviewing Editor and Reviewer #1, and the evaluation has been overseen by a Senior Editor.

Comments to the Authors:

We are sorry to say that, after consultation with the reviewers, we have decided that your work will not be considered further for publication by *eLife*.

The Cx43-mediated GJIC on HESCs would provide critical clues to understand the molecular etiology of endometriosis progression. However, all three reviewers pointed that the sample size of control and stage-specific endometriotic samples is too small for statistical analysis. Therefore, the results may be an artifact of individual differences in the patient samples that are not well controlled rather than due to endometriosis. Also, reviewers asked many questions regarding this manuscript because most of the results did not clearly support the authors' hypothesis.

*Reviewer #1 (Recommendations for the authors):*

The adhesion and invasion of shedding endometrial fragments made by retrograde menstruation to the target site are critical cellular processes to initiate endometriosis progression. However, the molecular etiology regarding the invasion process in endometriosis is not elucidated yet. This manuscript proposed that Cx43-mediated GJIC in HESCs is crucial to enhance adhesion and invasion processes of endometriosis.

Strength:

1. The activation of Cx43-mediated GJIC on HESCs would provide critical clues to understand the molecular etiology of endometriosis progression.

2. New sing cell-based gene expression profile with ESCs and ESCs between normal and endometriosis patients would provide the new information to define the molecular etiology of invasion and adhesion of endometriosis.

3. For the functional validation of the role of Cx43 in ESCs for the adhesion process in endometriosis, the author generates new Cx43 knockdown and overexpression of DN C43 mutant ESCs and employs a unique approach, such as heterotypic coupling assay and AFM assays.

Weakness:

1. The authors used only two human endometrial stromal cells from control, only one human endometrial stromal cells from stage I-II, and 5 endometrial stromal cells from stage III-IV endometriosis patients. The sample size of control and stage I-II is too small for statistical analysis compared to stage III-IV of endometriosis.

2. The authors' hypothesis may be opposite from their data. The authors claim that Cx43 mediated GJIP axis has an essential role in adhesion and invasion of ESCs from endometriosis patients compared to normal ESCs. However, the single-cell gene expression profile showed that ESCs from endometriosis patients have much less Cx43 levels than normal ESCs (Figure 1A and 1B). Also, Cx43 knockdown ESCs were generated from ESCs from endometriosis patients even though ESCs from endometriosis patients have low Cx43 levels. This discrepancy makes the reader confused.

3. In contrast with ESCs, EECs from endometriosis patients have higher levels of Cx43 than normal EECs. However, EECs from endometriosis patients did not have adhesion and invasion activity compared to control EECs. Therefore, it is not clearly described the cell-type-specific function of Cx43 in endometriosis progression.

There are main questions about the results need to be 'adequately addressed' to substantiate their conclusions.

1. The authors used only two human endometrial stromal cells from control, only one human endometrial stromal cells from stage I-II, and 5 endometrial stromal cells from stage III-IV endometriosis patients. The sample size of control and stage I-II is too small for statistical analysis compared to stage III-IV of endometriosis. Therefore, the authors should increase the number of human endometrial cells for the control and stage I-II.

2. In contrast with Figure 1A, the number of human endometrial epithelial cells from stage III-IV is only one in Figure 1B. Therefore, there is a problem in statistical analysis with only one sample for human endometrial epithelial cells from stage I-II and III-IV to determine the differential gene expression profile between different disease stages.

3. Most of the data in Figure 1E and 1F have no significant difference between control and endometriosis samples due to limited sample size. Therefore, it is hard to make a conclusion to support the author's hypothesis with these results. Also, 003 data set in Figure 1F is not shown in Figure 1B. Whys single-cell RNA analysis of human epithelial cells (003, NC) is not shown in Figure 1B.

4. In Figure 2, the percentage of SC-5 and EC-4 is inversely corrected with the percentage of SC-6 and EC-5 in an endometriosis stage-dependent manner. What is SC-5 and EC-4? Do these cell types have any molecular properties to enhance the endometriosis progression?

5. In Figure 3, the authors claimed that CX43 has an essential role in the GJIC of endometriotic ESCs compared to control ESCs. However, the authors showed that endometriotic ESCs have low levels of CX43 compared to control (Figure 1A). In Figure 3C, the authors showed that endometriotic ESCs have higher GJIC than normal ESCs even though endometriotic ESCs have lower Cx43 levels than control ESC. Previous studies revealed the promoting role of Cx43-GJIC in cell-cell adhesion and metastasis in prostate cancer, gastric cancer, and glioma cells.

Therefore, endometriotic ESCs (low Cx43) have higher GJIC than control ESC (high Cx43), which is the opposite of other studies. Thus, the authors should show Cx43 protein levels between control and endometriotic ESC by Western blotting. The authors should also determine loss- and gain-of Cx43 gene function in normal and endometriotic ESC and EECs to validate the role of CX43 in GJIC.

6. Figure 4, the authors claimed that endometriotic ESCs are the major invasive front into mesothelium cells. However, it is not clear whether control ESCs also attach to PMCs compared to the endometriotic ESCs. To validate Cx43 in cell-to-cell attachment, the loss- or gain-of Cx45 gene function in control and endometriotic ESC and EEC should be analyzed by AFM.

7. Cytokine, such as TNF α, induce invasion of refluxed endometrial tissues to enhance endometriosis progression. However, in Figure 5, there is no cytokine exposure to enhance invasion of endometrial cells from normal versus endometriosis patients. Therefore, there is no difference in invasion activity of ESC and EEC from normal versus endometriosis patients. Thus, the authors should conduct the cytokine-mediated invasion assay.

8. In Figure 5, the authors showed that invasion activity difference between normal versus endometriosis is detected when ESC was cocultured with EEC. In Figure 6, however, the authors used only ESCs to define the Cx45 knockdown effect on GJ changing and invasion activity in between normal versus endometriosis. Therefore, there is no significant difference in CX43 knockdown effect on GJ changing and invasion activity between control and endometriotic ESC. Therefore, these observations do not support the role of Cx43 in endometriosis progression. Therefore, the authors should employ the mixture of ESC and EEC to examine the difference between control versus endometriosis. In Figure 1, the authors showed that Cx levels are significantly redubbed in ESC from endometriosis than those in ESC from control. However, the authors did not show the differential levels of Cx43 in ESC between control versus endometriosis patients by Western blotting in this figure. If CX43 levels in endometriotic ESC are too low to be detected as compared with normal ESC, there is no meaning to define the Cx43 knockdown effect in endometriotic ESC. The gain-of Cx43 gene function study should be conducted in endometriotic ESC rather than Cx43 knockdown.

9. In Figure 7D, there is no significant difference in penetration value between control and endometriotic ESCs even though both ESCs elevated penetration value compared to no ESC control. This data implies that both control and endometriotic ESCs could disrupt a mesothelial cells' barrier function to initiate the endometriosis progression. Therefore, this result does not explain why about 10% of reproductive-aged women who have experienced retrograde menstruation have endometriosis.

*Reviewer #2 (Recommendations for the authors):*

The authors were trying to show how gap junction proteins are overexpressed, move to the cell membrane, and function to propel endometrial cells to invade into mesothelial cells.

The major strengths are the innovation of the techniques used. A weakness is the justification of the sample size and appropriate statistical analysis for each experiment.

The study achieved its aims as presented. However, there is some concern for rigor and reproducibility bases on sample size and statistical analyses.

This work, if reproducible, would have great impact on the field as endometriosis pathogenesis via retrograde menstruation remains largely unknown on a molecular and cellular level.

Phenotyping of endometriosis patients is critical. While ASRM staging criteria is a thing, more frequently indications for surgery such as pelvic pain or infertility are useful. Additionally, type of endometriosis (i.e., superficial peritoneal, ovarian endometrioma, rectovaginal nodules, iatrogenic, deep invasive endometriosis, or mixture) would be useful.

The methods mentions that one patient was on combined oral contraceptives but that is not mentioned in Table 1. Additionally, race and ethnicity should be shown and the method to determine both (i.e., patient described) should be described. Pregnancy status (current) or infertility or parity/gravidity should be provided for context.

The menstrual cycle phase has been shown to change in normal cycling women and to have unique features in women with endometriosis. It is unclear whether the menstrual cycle stage listed was consistent with last menstrual period, surgical pathology, or just based on steroid hormone levels.

Finally, the history of the control patients is critical. Please confirm that those women had no history of pelvic pain, endometriosis, or infertility. Please confirm that biopsies were done during surgery to prove that endometriosis was not present. 7% of women with pelvic pain have endometriosis on random biopsy without surgeon visualized lesion.

Ep-CAM as epithelial marker seems reasonable for normal eutopic epithelium. However, 12Z cells, an epithelial-like cell line derived from peritoneal lesions are E-cad negative yet N-cad positive. Please provide the appropriate rationale for using the Ep-CAM marker to discern epithelial cells. How different was the enrichment of EECs with Ep-CAM affinity column compared to differential attachment?

Please provide rationale for Actin as housekeeping gene when the assay contained other potential housekeeping genes. What is UBB?

The statistical analysis does not include a justification of sample size or power analysis.

Figure 1: the authors conclude that the gene expression patterns are different between normal and endometriosis via a heat map. In particular the statement that the transition to more aggressive disease is evident is not clearly visualized. In particular, this dataset needs to be objectively assessed using statistical means to support the conclusions.

It is interesting that the single cell analysis was then combined into violin plots.

The details of the figures are vague. There are blue arrows on figure 2 but I cannot tell from the figure legend what that means.

Figure 3 B – additional images showing similar results are needed. It is unclear why the parachute test was not performed with epithelium and stroma. I understand that the goal is to look at the invasion through mesothelium but as a control for proof of principal epithelial stromal communication should be shown. Or at least discussed why you believe this is a different mechanism.

Please justify the sample size for Figure 4.

Figure 6: do you have images of the DN Cx43 showing the localization of the protein that fails to function in cells?

Figure 7 is my favorite figure. I think you could provide a more descriptive cartoon of this. And highlight it on Figure 8.

*Reviewer #3 (Recommendations for the authors):*

The introduction could be substantially shortened.

The discussion includes what should be in the Results section.

It would be of interest to examine the endometriosis itself rather than the endometrial cells of women with the disease.

The data are well done however the use of a very small number of endometrial cells and not endometriosis precludes generalization. The results may be an artifact of individual differences in the patient samples that are not well controlled rather than due to endometriosis.

**[Editors’ note: further revisions were suggested prior to acceptance, as described below.]**

Thank you for resubmitting your work entitled "Hypersensitive intercellular responses of endometrial stromal cells drive invasion in Endometriosis" for further consideration by *eLife*. Your revised article has been evaluated by Diane Harper (Senior Editor) and a Reviewing Editor.

The manuscript has been improved but there are some remaining issues that need to be addressed, as outlined below:

Reviewer #1:

The authors are attempting to show how gap junction proteins are over-expressed in the endometrium of women with endometriosis, and is one mechanism by which endometriosis/the endometrium of women with endometriosis has invasive/migratory potential/ability. The strengths include their various methods utilized/described. They use their results to postulate that the invasiveness/migratory capabilities of ectopic endometrium from women with endometriosis is mediated by gap junctions, and that it provides further support for the role of the endometrium in allowing endometriosis to be an invasive disorder.

Recommendations for the authors:

Introduction:

It would have been helpful to have more information on the comparison/rationale for looking at gap junctions from background from cancer literature. Also, would recommend mentioning the role of stem cells as another potential etiology of endometriosis

Methods:

While the methods performed are robust, why wasn't histology used as a first line assessment of invasion performed as a first line?

The statistical analysis does not include a justification of sample size or power analysis-- was this performed?

Also, what is the rationale for some of the analyses performed as one sided t tests?

Results: an overall concern is the at times small sample size-- while the authors note in their responses an increased sample size, this is not consistently seen across all experiments.

If the gap junctions are over all low, how can we justify the great importance placed on CX43. Furthermore, prior studies have consistently commented on low Cx43, so I believe there is still some confusion for why Cx43 knockdown was performed if it is already low (Yu J., Boicea A., Barrett K.L., James C.O., Bagchi I.C., Bagchi M.K., Nezhat C., Sidell N., Taylor R.N. Reduced connexin 43 in eutopic endometrium and cultured endometrial stromal cells from subjects with endometriosis. Mol. Hum. Reprod. 2014;20:260-270. doi: 10.1093/molehr/gat087; Regidor P.A., Regidor M., Schindler A.E., Winterhager E. Aberrant expression pattern of gap junction connexins in endometriotic tissues. Mol. Hum. Reprod. 1997;3:375-381. doi: 10.1093/molehr/3.5.375;Winterhager E., Grummer R., Mavrogianis P.A., Jones C.J., Hastings J.M., Fazleabas A.T. Connexin expression pattern in the endometrium of baboons is influenced by hormonal changes and the presence of endometriotic lesions. Mol. Hum. Reprod. 2009;15:645-652. doi: 10.1093/molehr/gap060).

Specific questions based on figures/descriptions:

If force could not accurately be measured, then how did they get accurate measurements

For Figure 1C matrigel, it is said that PMC are needed to invade and that this is why not much invasion was seen with ESCs, but then more this is followed by more invasion was seen stage 3/4… not completely consistent

For Figure 1E, small sample size of control (only 2). Again, seem to be presenting contradicting information because initially it is noted that the focus is on ESCs, but then later noting requiring EECs with ESCs to help with invasion?

Figure 2

The argument appears to be that motility is higher in endometriosis cells, but it is not clear if the peritoneum is needed/necessary or not? Also the sample sizes remains small for PMCs mixed with ESCs which makes interpretation challenging.

Figure 3

Image B and C is hard to interpret/does not appear to be c/w their hypothesis of requiring PMCs.

Would prefer clearer descriptions for staining seen in D onward as it is not completely clear

For Results section entitled: GJIC is required for invasion of ESCs across a peritoneal mesothelium:

When you reference Chen et al. 2021, which ESCs are you referring to? Non endometriosis, because in the intro it is noted that Cx43 is lower in endo…?

For Figure 4a, were ESCs, EECs and PMCs combined for control vs endometriosis? This is not clear

Was n=3 for control and endo each? This is still a small sample size

siRNA data seems less reliable given the compromised cell health

Furthermore, why was siRNA/shRNA done when it was already noted that Cx43 is lower in endo and why is there lack of clarity with respect to siRNA targeting PMCs vs ESC?

Figure 4E results:

How was it ensured that loss of any possible adhesive roles of gap junctions was not responsible given that the GJIC could not be assessed? Also some of the sample sizes remained small

Figure 5

Small samples again (n=3) which was a prior critique. Also, how many control samples were used, it is not mentioned in the results

Figure 5F and Figure 5G

Results seem contradictory, with respect to hypothesis of Cx43

Figure 6

What were the sample sizes?

Discussion

Third to last paragraph seems contradictory to what is being described up until this point

Last paragraph: seems a bit of a jump to say that the changes start/ are primed in the endometrium and comparing the endometrium to a primary tumor.

Overall, the discussion appears to simply repeat the results, rather than providing additional insights based on the results

*Reviewer #2:*

The manuscript by Chen et al. on "Hypersensitive intercellular responses of endometrial stromal cells drive invasion in Endometriosis" is interesting. The authors show that ESCs isolated from patients with endometriosis are more invasive than EECs. ESCs induce gap junctions when they merge with peritoneal mesothelium that further enhance the function of ESCs. The manuscript provides new data on the invasiveness of ESCs into mesothelium while initiating lesion formation. However, there are some issues that need to be addressed.

Discussion:

– Authors are advised to remove figure numbers in the Discussion section. Instead, mention in the results.

– Discussion should be rewritten after removing figure numbers.

Figure Legends:

All figure legends are too lengthy. They should be rewritten in a simple way while avoiding unnecessary explanation in figure legends. Some of the text should go to the Methods section and remaining explain in the Results section.

Figure 1: Change the title to:

"Characterization of endometrial cells from patients with endometriosis"

Figure 4: Change the title to:

Invasiveness of ESCs Invasiveness is dependent on Cx43 GJIC

Figure 5: Change the title to:

"ESCs induce disruption of the barrier function of a mesothelial

monolayer"

Figure 6: Change the title to:

"Disruption of the mesothelial barrier by ESCs is propagated through

Cx43 gap junctions."

Figures:

Figure 2: Give a space between "A" panel and "D and F" panel.

Figure 3: Give a space between "A" panel and "B, C and D" panel. Labels B, C and

D are very close to the "a" panel image.

Figure 4: Labeling is not very clear: Authors should remove "Cx43 siRNA

transfection GJIC Invasion" and "Cx43 shRNA and DN infection GJIC" from

the figure. Instead, they can mention in Figure legend.

– Give space between upper panel and lower panel figures.

Figure 7: Label "B" is missing on the model figure.

**[Editors’ note: further revisions were suggested prior to acceptance, as described below.]**

Thank you for resubmitting your work entitled "Hypersensitive intercellular responses of endometrial stromal cells drive invasion in Endometriosis" for further consideration by *eLife*. Your revised article has been evaluated by Diane Harper (Senior Editor) and a Reviewing Editor.

The manuscript has been improved but there are some remaining issues that need to be addressed, as outlined below:

The manuscript is greatly improved. The differences between patents is a significant concern. Are they similar between cases and controls? The menstrual cycle stage, use of OCs and post-partum state are major influences on endometrial cell behavior.

The author should also consider the possibility that the changes seen in endometrial cells in women with endometriosis are not the cause of the endometriosis, but instead caused in response to endometriosis. In animal models many of the features of endometriosis are induced in eutopic endometrium by placing normal endometrial cells in the peritoneal cavity. Please discuss this alternative explanation.

Please also discuss endometriosis outside of the peritoneal cavity and the stem cell model. Similar to peritoneal endometriosis, all women have circulating stem cells, yet few get endometriosis. Do those stem cells similarly have the characteristics as described here?

---

## [Author Response]

**[Editors’ note: the authors resubmitted a revised version of the paper for consideration. What follows is the authors’ response to the first round of review.]**

Comments to the Authors:We are sorry to say that, after consultation with the reviewers, we have decided that your work will not be considered further for publication by eLife.The Cx43-mediated GJIC on HESCs would provide critical clues to understand the molecular etiology of endometriosis progression. However, all three reviewers pointed that the sample size of control and stage-specific endometriotic samples is too small for statistical analysis. Therefore, the results may be an artifact of individual differences in the patient samples that are not well controlled rather than due to endometriosis. Also, reviewers asked many questions regarding this manuscript because most of the results did not clearly support the authors' hypothesis.Reviewer #1 (Recommendations for the authors):The adhesion and invasion of shedding endometrial fragments made by retrograde menstruation to the target site are critical cellular processes to initiate endometriosis progression. However, the molecular etiology regarding the invasion process in endometriosis is not elucidated yet. This manuscript proposed that Cx43-mediated GJIC in HESCs is crucial to enhance adhesion and invasion processes of endometriosis.Strength:1. The activation of Cx43-mediated GJIC on HESCs would provide critical clues to understand the molecular etiology of endometriosis progression.2. New sing cell-based gene expression profile with ESCs and ESCs between normal and endometriosis patients would provide the new information to define the molecular etiology of invasion and adhesion of endometriosis.3. For the functional validation of the role of Cx43 in ESCs for the adhesion process in endometriosis, the author generates new Cx43 knockdown and overexpression of DN C43 mutant ESCs and employs a unique approach, such as heterotypic coupling assay and AFM assays.Weakness:1. The authors used only two human endometrial stromal cells from control, only one human endometrial stromal cells from stage I-II, and 5 endometrial stromal cells from stage III-IV endometriosis patients. The sample size of control and stage I-II is too small for statistical analysis compared to stage III-IV of endometriosis.

Since the original submission, we have taken two years to acquire a greatly expanded patient sample base (acquisition of patient volunteers is slow) and increased the number of samples analyzed. Thus, we feel the original concern at the earlier stage of this study has been resolved.

2. The authors' hypothesis may be opposite from their data. The authors claim that Cx43 mediated GJIP axis has an essential role in adhesion and invasion of ESCs from endometriosis patients compared to normal ESCs. However, the single-cell gene expression profile showed that ESCs from endometriosis patients have much less Cx43 levels than normal ESCs (Figure 1A and 1B). Also, Cx43 knockdown ESCs were generated from ESCs from endometriosis patients even though ESCs from endometriosis patients have low Cx43 levels. This discrepancy makes the reader confused.

Firstly, the single cell gene expression from our expanded patient base has now been submitted for publication separately to an Ob/Gyn specialty journal, allowing us to focus on cell mechanism and physiology here.

Secondly, the confusion of the reviewer is understandable, as most of the connexin genes we analyzed show decreased expression in endometriosis derived ESCs. However, the expression levels of Cx43, the major connexin (10x higher than other Cxs), does not change significantly with disease, consistent with the coupling levels we see. The major difference is the induction of coupling seen with endometriosis stromal cells when encountering the mesothelium due to trafficking rather than increased expression. We believe it is the change in expression level, from low to high on encountering the mesothelium, that triggers phenotypic changes in both cell lines.

3. In contrast with ESCs, EECs from endometriosis patients have higher levels of Cx43 than normal EECs. However, EECs from endometriosis patients did not have adhesion and invasion activity compared to control EECs. Therefore, it is not clearly described the cell-type-specific function of Cx43 in endometriosis progression.

We now better understand the role of EECs in endometriosis, in that they do not form the predominant invasive species but do seem to enhance invasiveness. As is the case for ESCs, while many minor Cx species increase expression levels with endometriosis, Cx43, which is most abundant and contributes most functional coupling, does not change greatly.

There are main questions about the results need to be 'adequately addressed' to substantiate their conclusions.1. The authors used only two human endometrial stromal cells from control, only one human endometrial stromal cells from stage I-II, and 5 endometrial stromal cells from stage III-IV endometriosis patients. The sample size of control and stage I-II is too small for statistical analysis compared to stage III-IV of endometriosis. Therefore, the authors should increase the number of human endometrial cells for the control and stage I-II.

As noted above, in the 2 years since this review, we have significantly expanded sample sizes so that statistical analysis is robust.

2. In contrast with Figure 1A, the number of human endometrial epithelial cells from stage III-IV is only one in Figure 1B. Therefore, there is a problem in statistical analysis with only one sample for human endometrial epithelial cells from stage I-II and III-IV to determine the differential gene expression profile between different disease stages.

Expression profiles have now been submitted separately, given the expanded data set and analysis. The current version of the manuscript focuses on mechanistic studies.

3. Most of the data in Figure 1E and 1F have no significant difference between control and endometriosis samples due to limited sample size. Therefore, it is hard to make a conclusion to support the author's hypothesis with these results. Also, 003 data set in Figure 1F is not shown in Figure 1B. Whys single-cell RNA analysis of human epithelial cells (003, NC) is not shown in Figure 1B.

See point 2 above.

4. In Figure 2, the percentage of SC-5 and EC-4 is inversely corrected with the percentage of SC-6 and EC-5 in an endometriosis stage-dependent manner. What is SC-5 and EC-4? Do these cell types have any molecular properties to enhance the endometriosis progression?

See point 2 above.

5. In Figure 3, the authors claimed that CX43 has an essential role in the GJIC of endometriotic ESCs compared to control ESCs. However, the authors showed that endometriotic ESCs have low levels of CX43 compared to control (Figure 1A). In Figure 3C, the authors showed that endometriotic ESCs have higher GJIC than normal ESCs even though endometriotic ESCs have lower Cx43 levels than control ESC. Previous studies revealed the promoting role of Cx43-GJIC in cell-cell adhesion and metastasis in prostate cancer, gastric cancer, and glioma cells.Therefore, endometriotic ESCs (low Cx43) have higher GJIC than control ESC (high Cx43), which is the opposite of other studies. Thus, the authors should show Cx43 protein levels between control and endometriotic ESC by Western blotting. The authors should also determine loss- and gain-of Cx43 gene function in normal and endometriotic ESC and EECs to validate the role of CX43 in GJIC.

While many Cx genes do show decreased expression in endometriosis, the major one, Cx43, which contributes most or all of the functional coupling, is not greatly changed in level with disease, with protein levels approximately consistent with the observed coupling of ESCs to one another.

6. Figure 4, the authors claimed that endometriotic ESCs are the major invasive front into mesothelium cells. However, it is not clear whether control ESCs also attach to PMCs compared to the endometriotic ESCs. To validate Cx43 in cell-to-cell attachment, the loss- or gain-of Cx45 gene function in control and endometriotic ESC and EEC should be analyzed by AFM.

This is an interesting point, and we have been trying to conduct these studies, but there have been technical challenges with the use of AFM for these measurements.

We do have preliminary data indicating the endometriosis ESCs adhere much more quickly, and much more strongly, to mesothelial cells, but as this is based on a minimal n. We do not feel it can be published at this point. We do not want to further delay publication of our extensive data sets pending this (the measurements will require a new instrument that will not arrive for another 5 months), as there are many other aspects that explain higher invasiveness in endometriosis (enhanced ESC motility, ESC-PMC coupling and responsiveness to EECs).

7. Cytokine, such as TNF α, induce invasion of refluxed endometrial tissues to enhance endometriosis progression. However, in Figure 5, there is no cytokine exposure to enhance invasion of endometrial cells from normal versus endometriosis patients. Therefore, there is no difference in invasion activity of ESC and EEC from normal versus endometriosis patients. Thus, the authors should conduct the cytokine-mediated invasion assay.

The suggestion of TNF α was insightful, and indeed it has now been reported that it specifically stimulates endometriosis stromal cells to produce the CXCL^-^16 cytokine, which in turn stimulates motility and invasion. We have tested this cytokine as a chemoattractant, but it does not show a difference between control and endometriosis derived ESCs. Instead, we have modified our invasion assays to provide a general chemoattractant gradient (0 to 1% serum). This does now show a dramatic difference in invasiveness of ESCs between control and endometriosis patients. This is now in the manuscript, and we want to extend our appreciation to the reviewer for their insight.

8. In Figure 5, the authors showed that invasion activity difference between normal versus endometriosis is detected when ESC was cocultured with EEC. In Figure 6, however, the authors used only ESCs to define the Cx45 knockdown effect on GJ changing and invasion activity in between normal versus endometriosis. Therefore, there is no significant difference in CX43 knockdown effect on GJ changing and invasion activity between control and endometriotic ESC. Therefore, these observations do not support the role of Cx43 in endometriosis progression. Therefore, the authors should employ the mixture of ESC and EEC to examine the difference between control versus endometriosis. In Figure 1, the authors showed that Cx levels are significantly redubbed in ESC from endometriosis than those in ESC from control. However, the authors did not show the differential levels of Cx43 in ESC between control versus endometriosis patients by Western blotting in this figure. If CX43 levels in endometriotic ESC are too low to be detected as compared with normal ESC, there is no meaning to define the Cx43 knockdown effect in endometriotic ESC. The gain-of Cx43 gene function study should be conducted in endometriotic ESC rather than Cx43 knockdown.

We believe there was a misunderstanding of the interpretation of these results. Firstly, as noted above, there are not large differences in expression of Cx43 itself between control and endometriosis derived ESCs. The KD studies were to demonstrate that invasion itself is dependent on Cx expression, not that this is the primary difference between control and endometriosis cells, although we do show that the induction of gap junction coupling between ESCs and PMCs is much higher in endometriosis. Other aspects of endometriosis derived ESCs, such as ability to suppress apoptosis and survive in the peritoneum, are major contributors to disease, and not tested here (this has been documented previously, and our design by-passes these effects in order to focus on invasion itself). The point here is to document that GJs are required for invasion, and thus represent a potential target for disease therapy in the future. This dependence on gap junctions has never been documented previously in endometriosis.

9. In Figure 7D, there is no significant difference in penetration value between control and endometriotic ESCs even though both ESCs elevated penetration value compared to no ESC control. This data implies that both control and endometriotic ESCs could disrupt a mesothelial cells' barrier function to initiate the endometriosis progression. Therefore, this result does not explain why about 10% of reproductive-aged women who have experienced retrograde menstruation have endometriosis.

In our expanded data set, and statistical analysis, there is a significant difference between penetration induced by Endometriosis ESCs compared to control.

Reviewer #2 (Recommendations for the authors):The authors were trying to show how gap junction proteins are overexpressed, move to the cell membrane, and function to propel endometrial cells to invade into mesothelial cells.The major strengths are the innovation of the techniques used. A weakness is the justification of the sample size and appropriate statistical analysis for each experiment.The study achieved its aims as presented. However, there is some concern for rigor and reproducibility bases on sample size and statistical analyses.This work, if reproducible, would have great impact on the field as endometriosis pathogenesis via retrograde menstruation remains largely unknown on a molecular and cellular level.

As noted above, we have addressed this major limitation of the prior submission by accumulating many more patient samples, and now demonstrate robust statistical differences between the endometriosis and control patients are indeed reproducible in terms of enhanced motility, coupling with mesothelial cells, and invasion.

Phenotyping of endometriosis patients is critical. While ASRM staging criteria is a thing, more frequently indications for surgery such as pelvic pain or infertility are useful. Additionally, type of endometriosis (i.e., superficial peritoneal, ovarian endometrioma, rectovaginal nodules, iatrogenic, deep invasive endometriosis, or mixture) would be useful.The methods mentions that one patient was on combined oral contraceptives but that is not mentioned in Table 1. Additionally, race and ethnicity should be shown and the method to determine both (i.e., patient described) should be described. Pregnancy status (current) or infertility or parity/gravidity should be provided for context.The menstrual cycle phase has been shown to change in normal cycling women and to have unique features in women with endometriosis. It is unclear whether the menstrual cycle stage listed was consistent with last menstrual period, surgical pathology, or just based on steroid hormone levels.Finally, the history of the control patients is critical. Please confirm that those women had no history of pelvic pain, endometriosis, or infertility. Please confirm that biopsies were done during surgery to prove that endometriosis was not present. 7% of women with pelvic pain have endometriosis on random biopsy without surgeon visualized lesion.

The detailed patient data for all patients used in the current study, including most of the parameters mentioned by the reviewer, are now presented in Table S1. Absence of endometriosis was confirmed surgically in all control patients by macroscopic examination, and where indicated by standard clinical practice, were confirmed by histological exam.

Ep-CAM as epithelial marker seems reasonable for normal eutopic epithelium. However, 12Z cells, an epithelial-like cell line derived from peritoneal lesions are E-cad negative yet N-cad positive. Please provide the appropriate rationale for using the Ep-CAM marker to discern epithelial cells. How different was the enrichment of EECs with Ep-CAM affinity column compared to differential attachment?

We separate EECs and ESCs by physical means (differential sedimentation and adhesion), and only used immunocytochemistry to confirm identity. We have mostly used Cytokeratin 7 (for EECs) and Vimentin (for ESCs), which has proven to be quite robust, in part because Ep-CAM is one of the proteins that change with endometriosis.

Please provide rationale for Actin as housekeeping gene when the assay contained other potential housekeeping genes. What is UBB?The statistical analysis does not include a justification of sample size or power analysis.Figure 1: the authors conclude that the gene expression patterns are different between normal and endometriosis via a heat map. In particular the statement that the transition to more aggressive disease is evident is not clearly visualized. In particular, this dataset needs to be objectively assessed using statistical means to support the conclusions.It is interesting that the single cell analysis was then combined into violin plots.The details of the figures are vague. There are blue arrows on figure 2 but I cannot tell from the figure legend what that means.

We have moved the single cell profiling aspect of the study to a separate submission focused on its potential as a diagnostic. This resubmitted manuscript now focuses exclusively on the cellular mechanisms underlying endometriosis, and the previously unexplored role of gap junctions in lesion formation

Figure 3 B – additional images showing similar results are needed. It is unclear why the parachute test was not performed with epithelium and stroma. I understand that the goal is to look at the invasion through mesothelium but as a control for proof of principal epithelial stromal communication should be shown. Or at least discussed why you believe this is a different mechanism.

The parachute assay works well with adherent cells, particularly when they can form a monolayer, which epithelial cells do not (they grow in small compact colonies, and do not plate down with good efficiency). This has made accumulation of statistically significant amounts of data on their coupling difficult. We have done some studies that confirm they couple to stromal cells, but not enough to conclude if this differs between endometriosis and control samples.

Figure 6: do you have images of the DN Cx43 showing the localization of the protein that fails to function in cells?

We do not have this, as the DN construct is not tagged and hence is indistinguishable from endogenous Cx43. However, there are images of this DN in the literature, including our original publication describing it referenced here, where it is shown to form completely normal gap junctions between cells at the EM level at much higher resolution that is provided by light microscopy.

Figure 7 is my favorite figure. I think you could provide a more descriptive cartoon of this. And highlight it on Figure 8.

We appreciate this comment, as do my AFM collaborators!! At the referees’ suggestion, we have now provided a more descriptive model in the text.

Reviewer #3 (Recommendations for the authors):The introduction could be substantially shortened.The discussion includes what should be in the Results section.It would be of interest to examine the endometriosis itself rather than the endometrial cells of women with the disease.

We have reduced the introduction, but it remains substantial as we have found that the broader cell and molecular biology audience is not very familiar with this disease, its burden and potential origins, so significant background is needed. We have moved all direct description of results to Results and left broader significance issues to Discussion.

The intent of the work is to investigate the early formation stages of endometrial lesions, and not the later development of the lesion, so the relevant cell types are the endometrial cells, in the same way that the relevant cells to study in the earliest stages of metastasis are primary tumor cells or circulating cancer stem cells, not the cells already in a metastatic lesion.

The data are well done however the use of a very small number of endometrial cells and not endometriosis precludes generalization. The results may be an artifact of individual differences in the patient samples that are not well controlled rather than due to endometriosis.

As noted above, we have now taken two years to significantly increase our patient database, and testing of many more control, and endometriosis samples (both stages I-II and III-IV), and through rigorous testing, have supported our initial conclusions.

**[Editors’ note: what follows is the authors’ response to the second round of review.]**

The manuscript has been improved but there are some remaining issues that need to be addressed, as outlined below:Reviewer #1:The authors are attempting to show how gap junction proteins are over-expressed in the endometrium of women with endometriosis, and is one mechanism by which endometriosis/the endometrium of women with endometriosis has invasive/migratory potential/ability. The strengths include their various methods utilized/described. They use their results to postulate that the invasiveness/migratory capabilities of ectopic endometrium from women with endometriosis is mediated by gap junctions, and that it provides further support for the role of the endometrium in allowing endometriosis to be an invasive disorder.Recommendations for the authors:Introduction:It would have been helpful to have more information on the comparison/rationale for looking at gap junctions from background from cancer literature.

We had provided extensive citations on this, but have now expanded to include some mechanistic insights. Of particular relevance to the current study we note that in metastatic breast cancer there is enhanced trafficking of Cx43 to cell interfaces resulting in increased coupling [Kanczuga-Koda et al., 2006], as we show now in endometriosis (Figure 3)

Also, would recommend mentioning the role of stem cells as another potential etiology of endometriosis

Thank you for the suggestion. This and more specifics about these alternate models are now included.

Methods:While the methods performed are robust, why wasn't histology used as a first line assessment of invasion performed as a first line?

For our specific study, counting provides a quantitative measure of invasion. Histology is in some ways more complex, and detailed histological images have already been previously published in the cited papers [e.g. Nair et al., 2008]. The AFM studies were well adapted to measuring adhesion, and specifically mapping the role of GJs in mesothelial breakdown.

The statistical analysis does not include a justification of sample size or power analysis-- was this performed?

We did not perform initial Power Analyses, as we had no precedent ion which to base the likely size of any differences we would see between control and patient samples. Post-hoc power analysis did not seem like it would add to the rigor of the statistical tests we have applied. However, we have addressed this concern related to “n” by significantly increasing sample sizes in most of the experiments now presented (see below)_

Also, what is the rationale for some of the analyses performed as one sided t tests?

Thank you for noticing this….all T-tests are now 2-sided, which is appropriate, as noted by the reviewer.

Results: an overall concern is the at times small sample size-- while the authors note in their responses an increased sample size, this is not consistently seen across all experiments.

While all of these analyses are quite complex, making it challenging to accrue huge sample sets, we have now significantly increased the sample size in almost all experiments, increasing our overall patient samples to 22 controls and 22 endometriosis (equally divided between stages I-II and III-IV). We discuss each case below.

If the gap junctions are over all low, how can we justify the great importance placed on CX43. Furthermore, prior studies have consistently commented on low Cx43, so I believe there is still some confusion for why Cx43 knockdown was performed if it is already low (Yu J., Boicea A., Barrett K.L., James C.O., Bagchi I.C., Bagchi M.K., Nezhat C., Sidell N., Taylor R.N. Reduced connexin 43 in eutopic endometrium and cultured endometrial stromal cells from subjects with endometriosis. Mol. Hum. Reprod. 2014;20:260-270. doi: 10.1093/molehr/gat087; Regidor P.A., Regidor M., Schindler A.E., Winterhager E. Aberrant expression pattern of gap junction connexins in endometriotic tissues. Mol. Hum. Reprod. 1997;3:375-381. doi: 10.1093/molehr/3.5.375;Winterhager E., Grummer R., Mavrogianis P.A., Jones C.J., Hastings J.M., Fazleabas A.T. Connexin expression pattern in the endometrium of baboons is influenced by hormonal changes and the presence of endometriotic lesions. Mol. Hum. Reprod. 2009;15:645-652. doi: 10.1093/molehr/gap060)

We have tried to explain this better is the current version of the MS (including a closing sentence in the Introduction). Indeed, we do see a drop in Cx43 at the RNA level (presented in another manuscript under review) and functional level (Figure 3B), although not to the extent reported in Bagchi et al. above (note that Regidor looked at ectopic lesion expression that we do not address here). Most cells make more connexins than are required to achieve adequate GJ coupling of cells, so its distribution in the cell is critical. The point in this case is that much of the Cx43 that is expressed in endometrial cells is left in internal pools. In endometriosis patient ESCs, it is induced to go to the surface when they contact a mesothelial cell, enhancing functional GJ coupling more than seen in control cells (Figure 3C). Like many biological systems. It is not the absolute levels, but the change in level that is sensed by cells, which we propose occurs here. We would love to have a tool that specifically impacts only the GJ induction, but failing that the best way to test if this induction of GJ coupling between ESCs and PMCs is critical for invasion is overall Cx43 KD.

Specific questions based on figures/descriptions:If force could not accurately be measured, then how did they get accurate measurements

We show the accurate measurements for all control samples. The issue was that the endometriosis ESCs are clearly much more adhesive than control ESCs, as they form attachments much faster that are beyond the range of the instrument. Hence, we cannot say HOW MUCH more adhesive they are, but they are more adhesive than controls.

For Figure 1C matrigel, it is said that PMC are needed to invade and that this is why not much invasion was seen with ESCs, but then more this is followed by more invasion was seen stage 3/4… not completely consistent

These are two independent statements – ESCs do not invade across Matrigel in our 24 hour time-frame unless PMCs are present, indicating a need for PMCs to promote invasive behavior. But when PMCs are present, ESCs from endo patients, particularly stage III-IV, are much more invasive (Figure 1D left set of plats). We now add additional data and n from studies of primary PMCs that also support more invasion of Endo ESCs than controls, independent of the origin of the PMCs (i.e. control or endo patients). This provides an indication that disease is driven by the endometrium, rather than the peritoneum, but this is expanded on in a paper in press in Fert and Sterility Science,

For Figure 1E, small sample size of control (only 2). Again, seem to be presenting contradicting information because initially it is noted that the focus is on ESCs, but then later noting requiring EECs with ESCs to help with invasion?

Firstly, we have significantly expanded the n for these experiments (>8 for all cases). Secondly, we have now inserted a new Figure 1D to directly document the lower invasion rate of EECs compared to ESCs, as we expected from their lower adhesion. This strengthens the rationale to mostly focus on ESCs, However, since BOTH ESCs and EECs are present in vivo in lesions, we felt ignoring them would open us up to the valid criticism of not using physiologically relevant conditions. Figure 1F is important as it shows that EECs can further promote ESC invasiveness, but only to a significant level in endometriosis patients. We mention in the text that ESCs always remain the dominant invasive species, with EECs constituting 20 and 40% in controls and endo samples. This observation reinforces the major tenant of the MS title, that endometriosis ESCs are hypersensitive to the influence of other cells.

NOTE: In order to maintain clarity of which plots reflect ONLY ESCs, and which co-cultures, we have used color coding, with black and grey for control and endo ESCs, green for EECs and Red for PMCs, so graphs reflecting co-cultures show bi-colors.

Figure 2The argument appears to be that motility is higher in endometriosis cells, but it is not clear if the peritoneum is needed/necessary or not? Also the sample sizes remains small for PMCs mixed with ESCs which makes interpretation challenging.

ESCs alone are 2 fold more motile from endometriosis than control patients as shown in Figure 2C (only ESCs used in this experiment). Figure 2D then also shows that PMCs can further enhance motility of endometriosis ESCs by 1.5 fold, but have no significant effect on controls. The net effect is, in the presence of PMCs, endometriosis ESCs are 3 fold more motile. We have also addressed the “n”, with robust numbers for both single cultures and co-cultures (N>13 for all cases, with 4-15 patients in each).

Figure 3Image B and C is hard to interpret/does not appear to be c/w their hypothesis of requiring PMCs.

This is an example where we feel the color coding mentioned above may help, as well as presenting the homocellular cultures of ESCs in one figure (B) and the heterocellular cultures in a different figure (C). Hopefully this now shows that while ESC-ESC coupling actually drops by about 30% in endometriosis (as noted above, consistent with the direction if not magnitude reported by others), what is more critical is that ESC-PMC coupling is induced to a much greater extent in endometriosis.

Would prefer clearer descriptions for staining seen in D onward as it is not completely clear

While trying to keep the legend as brief as possible, based on Reviewer 2 comments, we have tried to clarify what we admit is a complex figure. The co-cultures (right panels) show a higher incidence of punctate staining at the interface of cells (arrowheads) than ESCs alone (left panels) indicative of better trafficking of Cx43 to the surface. However, this is perhaps best appreciated by the decrease in staining of intracellular pools in the co-cultures.

For Results section entitled: GJIC is required for invasion of ESCs across a peritoneal mesothelium:When you reference Chen et al. 2021, which ESCs are you referring to? Non endometriosis, because in the intro it is noted that Cx43 is lower in endo…?

Chen et al. looked at both control and endo ESCs, and observed that ALL Cx expression (of the 14 studied) drops in endometriosis ESCs (although it increases in EECs), but Cx43 remains by far the most dominant (10X higher in PCR), and actually decreases by much less than the other Cxs only by about 35%, reflective of the drop in coupling we show in Figure 3B.

For Figure 4a, were ESCs, EECs and PMCs combined for control vs endometriosis? This is not clearWas n=3 for control and endo each? This is still a small sample size

No EECs were in any of the experiments shown in Figure 4. In Figure 4A, BOTH PMCs and ESCs were exposed to GAP27 prior to invasion, as we have found that this is the only way for GAP27 to effectively block new GJ formation. The reason why this sample size has remained low, is that, as noted in the text, this was a much less consistent method of blocking GJIC, we suspect due to variability in the quality of different commercial batches of the peptide. Hence we focused initially on siRNA, before finally moving to virally expressed shRNAs.

siRNA data seems less reliable given the compromised cell health

We (unwisely) had tried to present all of our data despite the issues we had realized with transient siRNA transfection compromising the invasive ability of ESCs or ability of PMCs to form a stable barrier. Thus, we have gratefully followed the reviewer’s advice and eliminated the siRNA studies, and focused on shRNA. We did leave the GAP17 data in, despite its limits, as at least it represents a completely independent method of GJ block.

Furthermore, why was siRNA/shRNA done when it was already noted that Cx43 is lower in endo and why is there lack of clarity with respect to siRNA targeting PMCs vs ESC?

As noted above, endometriosis ESCs still have 60+% of the protein and coupling seen in controls, which is significant, and upon interacting with PMCs, coupling of endometriosis ESCs is induced such that they actually couple better than controls. We also believe it is this change in coupling level that induces so much change in the endo ESCs (again, the “hypersensitivity” theme), but without the ability to selectively ablate the induction, we have used the shRNA KD approach.

Figure 4E results:How was it ensured that loss of any possible adhesive roles of gap junctions was not responsible given that the GJIC could not be assessed? Also some of the sample sizes remained small

We can rule out adhesive roles of GJs as the DN T134A mutant forms robust GJ structures with all adhesive functions intact, but still prevent invasion. As we were the lab that developed this tool (Beahm et al., 2006), we have tested this mutant in MANY cell systems to document its block of coupling. These are complex experiments with primary cells, so the n is relatively low, but the strength of the conclusions lies in how three independent methods of blocking GJIC (4 if you count the more limited siRNA strategy) all yield the same effect.

Figure 5Small samples again (n=3) which was a prior critique. Also, how many control samples were used, it is not mentioned in the results

We had included the dye leak experiment as a means of measuring barrier breakdown of the PMCs based on the suggestion of a prior reviewer. We thank this reviewer for challenging the number of repeats, so we have now performed a number of parallel studies using dye leak and invasion measurements and have found they do NOT correlate. It appears that there the holes opened in the PMC monolayer by ESCs are very quickly “plugged” by the ESCs invading through them (reminiscent of the Dutch boy and the Dyke) so that significant dye leak is prevented Most prior studies using dye leak did not include invading cells. So we have eliminated this panel.

Figure 5F and Figure 5GResults seem contradictory, with respect to hypothesis of Cx43

It is understandable how this result causes confusion, as we had not expected it either!! However, the result in E (prior panel F) actually makes the results in panel F (prior G) even more compelling. We have tried to explain this better in the text.

Figure 5E: Within an epithelium, or in this case a mesothelium, GJs have been shown to help integrate and reinforce the junctional contact between cells, and hence the overall integrity of its barrier function. Some of this may be via channels, but it is likely a lot of it is the formation of a larger junctional complex with other junctions (e.g. tight and adhesions junctions). The results here are consistent with this in the mesothelium.

Figure 5F: When ESCs contact PMCs, they form heterocellular GJs, and these apparently pass signals (yet to be defined) to promote breakdown of the barrier function. Even worse, as we show in Figure 6, these disruptive signals are then propagated through the PMC monolayer by the same GJs that were previously helping to stabilize the barrier function. So when ESCs are around, the effects of KD or overexpression REVERSE from what they were with PMCs alone.

Figure 6What were the sample sizes?

In this figure, we are showing two individual experiments to prove a principle, not establish the significance of an effect. Each experimental condition is complex, as is each data point collected. Also, the dynamics of the culture system are such that experiments would not readily superimpose, and “average”. But each experiment in panels C and D illustrates the principle that signals that disrupt the mesothelial barrier are transmitted within the mesothelium by gap junctions.

DiscussionThird to last paragraph seems contradictory to what is being described up until this pointLast paragraph: seems a bit of a jump to say that the changes start/ are primed in the endometrium and comparing the endometrium to a primary tumor.Overall, the discussion appears to simply repeat the results, rather than providing additional insights based on the resultsReviewer #2:The manuscript by Chen et al. on "Hypersensitive intercellular responses of endometrial stromal cells drive invasion in Endometriosis" is interesting. The authors show that ESCs isolated from patients with endometriosis are more invasive than EECs. ESCs induce gap junctions when they merge with peritoneal mesothelium that further enhance the function of ESCs. The manuscript provides new data on the invasiveness of ESCs into mesothelium while initiating lesion formation. However, there are some issues that need to be addressed.Discussion:– Authors are advised to remove figure numbers in the Discussion section. Instead, mention in the results.– Discussion should be rewritten after removing figure numbers.

These corrections have been made.

Figure Legends:All figure legends are too lengthy. They should be rewritten in a simple way while avoiding unnecessary explanation in figure legends. Some of the text should go to the Methods section and remaining explain in the Results section.

We have made an effort to reduce Figure legends, particularly in relation to methods unless needed to interpret the figure. However, we believe there is value in the legend summarizing the result in the figure, as casual readers do focus on the figures rather than the text, yet they still need to understand the findings. So while we have abbreviated summaries of the results, they remain in the legends.

Figure 1: Change the title to:"Characterization of endometrial cells from patients with endometriosis"Figure 4: Change the title to:Invasiveness of ESCs Invasiveness is dependent on Cx43 GJICFigure 5: Change the title to:"ESCs induce disruption of the barrier function of a mesothelialmonolayer"Figure 6: Change the title to:"Disruption of the mesothelial barrier by ESCs is propagated throughCx43 gap junctions."

All titles have been changed as suggested. We appreciate the detailed and constructive suggestion!

Figures:Figure 2: Give a space between "A" panel and "D and F" panel.

Done – thank you

Figure 3: Give a space between "A" panel and "B, C and D" panel. Labels B, C andD are very close to the "a" panel image.

Done – thank you

Figure 4: Labeling is not very clear: Authors should remove "Cx43 siRNAtransfection GJIC Invasion" and "Cx43 shRNA and DN infection GJIC" fromthe figure. Instead, they can mention in Figure legend.

Based on suggestion from Reviewer 1, siRNA data has been removed. We have also removed all titles as suggested

– Give space between upper panel and lower panel figures.Figure 7: Label "B" is missing on the model figure.

Thank you…this oversight has been corrected

**[Editors’ note: what follows is the authors’ response to the third round of review.]**

The manuscript has been improved but there are some remaining issues that need to be addressed, as outlined below:The manuscript is greatly improved. The differences between patents is a significant concern. Are they similar between cases and controls? The menstrual cycle stage, use of OCs and post-partum state are major influences on endometrial cell behavior.

We now address this in the Discussion, p. 9, paragraph 3. While it is inevitable that the numbers of patients in each group at various stages of the menstrual cycle or on OCPs varied, we went back and analyzed the data segregating patients into those collected in the Secretory or Proliferative stages of the menstrual cycle or those on OCPs. While numbers in each group were somewhat limited, in every group invasion, migration and induction of GJIC were higher in Endometriosis derived samples. In addition, there were enough samples from Endometriosis patients to compare absolute levels of invasiveness, migration and induction of GJIC among the different groups. This revealed no statistically significant differences between groups. There was a trend for higher levels of invasion in patients on OCP, but this was more likely due to all of these patients being in the Endo III-IV group. So while we agree that hormone levels are likely to affect many aspects of endometrial behavior, this seems minimal with regard to the invasive characteristics, and is certainly overshadowed by differences between control and disease patients. We have not added these analyses as data to the paper, Author response image 1.

**Author response image 1. sa2fig1:** 

The author should also consider the possibility that the changes seen in endometrial cells in women with endometriosis are not the cause of the endometriosis, but instead caused in response to endometriosis. In animal models many of the features of endometriosis are induced in eutopic endometrium by placing normal endometrial cells in the peritoneal cavity. Please discuss this alternative explanation.

We now address this issue directly in Paragraph 4 on p. 9 in the Discussion, even adding a new reference that does suggest endometriosis can progress to more severe forms on regression. As we cannot collect tissue from patients who do not have endometriosis but develop it later, it is not possible here to directly test this possibility. However, all the changes we report here in endometriosis patients imbue endometrial cells with a more invasive phenotype, a required step for lesion formation. As this phenotype is NOT influenced differently by mesothelial cells from controls or patients (see Go et al., 2024), Occam’s razor would lead one to conclude that the endometrium is playing the key role in the origin of the disease because of these changes, even if it these could change further in disease progression.

Please also discuss endometriosis outside of the peritoneal cavity and the stem cell model. Similar to peritoneal endometriosis, all women have circulating stem cells, yet few get endometriosis. Do those stem cells similarly have the characteristics as described here?

The potential role of Stem cells in the endometrium is now mentioned at the end of para. 1 on p. 9 in the Discussion. It seems likely that any stem cells might differentiate in culture, but in any event they would make up a small enough fraction of the cells that they would not significantly influence our results. Assessing the properties of circulating stem cells is clearly beyond the scope of what is an already very extensive study of the endometrium. However, we had also been intrigued as to how the relatively rare occurrence of endometriosis lesions outside the peritoneum occurs. So we tested the invasion of ESCs across HUVECs as an indication of whether they might directly enter the bloodstream. In a comparison of 8 patients with significantly different invasive capability across PMCs, we found this to correlate very closely with invasion across HUVECs (R^2^ of 0.97). This data has now been included ad Figure 1G. As explained in para 1 of p.10 in the Discussion, this supports the idea that rather than invoking circulating stem cells as the origin of extra-peritoneal lesions, which certainly remains possible, it is likely that ESCs enter the circulation and given their suppressed apoptosis, as shown by others, could well lead to extra-peritoneal lesions.